# Training a high-performance retinal foundation model with half-the-data and 400 times less compute

Justin Engelmann[1,2,3] ✉ & Miguel O. Bernabeu ⓘ [1]

Medical artificial intelligence is limited by available training datasets. Foundation models like RETFound from Moorfields Eye Hospital (MEH) can be adapted with small downstream datasets and thus alleviate this issue. RETFound-MEH used 900,000 training images. Recently, "data-efficient" DERETFound achieved comparable performance with 150,000 images. Both require very substantial compute resources for training and use. We propose RETFound-Green trained on only 75,000 publicly available images with 400 times less compute using a novel Token Reconstruction objective. RETFound-MEH and DERETFound training costs are estimated at $10,000 and $14,000, respectively. RETFound-Green cost less than $100, with equally reduced environmental impact. RETFound-Green can be downloaded 14 times faster, computes vector embeddings 2.7 times faster which then require 2.6 times less storage space. On a variety of downstream tasks from geographically diverse datasets, RETFound-Green achieves more than twice as many statistically significant wins than the next best model.

Artificial Intelligence has many promising applications in medicine, but the lack of large labelled datasets and access to vast computational resources present substantial bottlenecks[1]. Domain-specific "foundation models" that can be efficiently adapted to various downstream tasks are being proposed to remedy this issue[2–4]. Zhou et al.[5] from Moorfields Eye Hospital (MEH) recently proposed such a foundation model for retinal imaging, called "RETFound". This could help unlock the potential of artificial intelligence in ophthalmology, where low-cost, non-invasive retinal colour fundus images are routinely used to screen for and diagnose retinal disease, a key public health burden[6,7]. Artificial intelligence could aid in the interpretation of these images[8–13]. These images capture the retina and its blood vessels in detail and thus allow inferences about the systemic health of individuals[14–17], too, a field of study known as "oculomics"[18].

RETFound, henceforth referred to as "RETFound-MEH" for Moorfields Eye Hospital, presents a great contribution to the field, but took substantial resources to train: 900,000 retinal colour fundus images that are largely not publicly available and two weeks of eight high-end A100 datacentre-grade GPUs, which we estimate to have cost

over $10,000 (see Methods for calculations). Note that this is only for the training the final model and not additional experimentation that is typically necessary when training deep learning models. This level of resource consumption makes further scaling up expensive, both financially and environmentally[19,20], and puts foundation model development out of reach of all but the most well-resourced labs.

More recently, Yen et al. proposed a data-efficient "DERETFound"[21] that was trained using only 150,000 images that are all publicly available. To accomplish this, they trained a diffusion model to generate over a million synthetic colour fundus images and then trained their model first on the synthetic and then on the real images. However, while it lowers the bar for dataset size, it does so at substantially increased computational cost due to training the generator and then using it to generate images. Overall, DERETFound required about 45% more computational resources than RETFound-MEH.

Both RETFound-MEH and DERETFound use the "Masked Auto-Encoder" (MAE)[22] self-supervised learning strategy including the original hyperparameters, which Zhou et al. found to be more effective than other self-supervised strategies like SimCLR[23], SwAV[24], DINO[25] or

[1]Centre for Medical Informatics, Usher Institute, University of Edinburgh, Edinburgh, UK. [2]School of Informatics, University of Edinburgh, Edinburgh, UK. [3]Institute of Ophthalmology, University College London, London, UK. ✉e-mail: j.engelmann@ucl.ac.uk

MoCo-v3[26]. However, these strategies are proposed in the general computer vision literature and thus designed for a task that differs substantially from the development of retinal image foundation model. First, they train models from scratch using randomly initialised weights, which is particularly challenging and computationally expensive for modern vision transformer architectures[27] that do not have strong inductive biases like traditional convolutional neural networks. Both RETFound-MEH and DERETFound use these pre-trained weights and thus only need to adapt the model to retinal imaging. Second, general computer vision uses natural images which have a different structure and more high-level diversity. In a dataset like ImageNet, an image of a dog is very different from an image of a car, which both in turn are very different from an image of a plate of food. In ophthalmology, a healthy eye and an eye with age-related macular degeneration might only differ through the presence of small deposits called drusen, which show up as tiny specks on the images. Third, datasets in general computer vision are very large with tens of millions[28] or even billions of images[29].

Design choices common in computer vision that were adopted by both RETFound-MEH and DERETFound include the MAE self-supervised learning approach, as well as a low resolution of 224 by 224 pixels, a fraction of the 3–4000 pixels that modern retinal cameras tend to acquire, and using the "large" variant of the vision transformer architecture[27], which makes them somewhat computationally expensive during inference, i.e. when processing images with the model for downstream tasks. While training costs are large, if models are adopted in routine care, the inference costs are recurring and thus a substantial factor with a non-negligible environmental footprint[19]. MAE involves pixel-level re-construction of images which is effective for the high-level diversity in natural images, but might not be well-suited for capturing small structures.

In this work, we propose a novel Token Reconstruction self-supervised learning strategy that focuses on higher-level, abstract features and is designed for making domain-specific foundation models instead of training models from scratch for general computer vision. See the Methods section for a detailed explanation. We use our Token Reconstruction objective to train RETFound-Green, a high-performance retinal foundation model. Figure 1 gives an overview of how our model compares with RETFound-MEH and DERETFound. Our strategy allows RETFound-Green to be trained far more efficiently in terms of data and compute, and yields a model that is substantially more lightweight in downstream applications while not being systematically worse than the previous models. On the contrary, in our experiments, RETFound-Green achieves more than twice as many statistically significant wins as the next best model on a variety of downstream tasks on six datasets. We even find that RETFound-Green with simple linear probing achieves comparable performance to a fully finetuned version of RETFound-MEH. All of our experiments use open data and the methodology underlying each of the comparisons is explained in detail in the Methods section. We make RETFound-Green openly available and expect that it will not only be a useful tool for researchers in ophthalmic AI but democratise both access to and development of foundation models. Our Token Reconstruction objective is not explicitly designed for retinal image analysis and thus might find application in other medical and non-medical domains.

## Results

### Overview

RETFound-Green uses a novel Token Reconstruction self-supervised learning pre-training objective that allows it to be trained far more efficiently than RETFound-MEH and DERETFound which use the Masked Autoencoder (MAE)[22] objective. As shown in Table 1, this allows for substantial improvements in efficiency while providing excellent performance.

The following sections provide more detailed explanations of our results: the efficiency of our model in terms of training resources as well as downstream use, adapting the model feature vector embeddings for classification with logistic regression ("linear probing"), the quality of unsupervised two-dimensional projections of the data, external transportability for diabetic retinopathy from one dataset to another across very different populations and settings, and comparing RETFound-MEH with full finetuning to RETFound-Green with only linear probing. Supplementary S8 further examines the effectiveness of our Token Reconstruction objective by providing an ablation of RETFound-Green trained at the same lower resolution as the other two models. This version of RETFound-Green likewise achieves strong results, suggesting that our approach indeed is effective even without increasing the resolution.

### Training and downstream use efficiency

RETFound-Green was trained with only 75,000 images, half of what DERETFound was trained on and 12 times less than the original

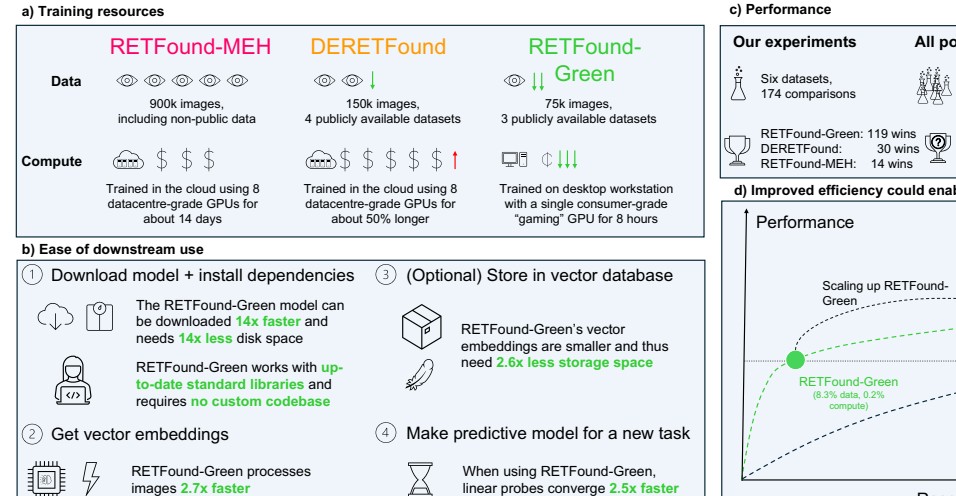

**Fig. 1 | Comparison of RETFound-MEH, DERETFound, and RETFound-Green.**
**a** RETFound-Green was trained with substantially less data and compute.
**b** RETFound-Green is easier and more efficient in downstream applications.
**c** RETFound-Green does not perform generally worse and better in most cases in our experiments, but this might differ across datasets and tasks. **d** RETFound-Green is a far more efficient approach and could be scaled up to yield an even higher performance, next-gen foundation model. GPU graphics processing unit.

**Table 1 | Training resources, downstream efficiency, and performance**

|  | RETFound-MEH | DERETFound | RETFound-Green | Green vs best |
|---|---|---|---|---|
| Training data | 904,170 images | 150,786 images | 75,000 images | 2× less |
| Training compute | 112 A100 days | 163 A100 days | ~0.27 A100 days | >400× less |
| Training cost (monetary/carbon, estimate) | ~$10,000/81 kg of coal burned | ~$14,000/117 kg of coal burned | <$100/0.2 kg of coal burned | >100× less |
| Training hardware | 8× top datacentre GPUs (total VRAM: 320GB) | 8× top datacentre GPUs (total VRAM: 640GB) | 1× top consumer gaming GPU (total VRAM: 24GB) | >8× less |
| Disk space (model) | 1.12GB (our optimisation, 3.68GB originally) | 1.12GB (our optimisation, 3.68GB originally) | 0.09GB | 14× less |
| Disk space (1 million embeddings) | 39.1GB | 39.1GB | 14.6GB | 2.6× less |
| Inference speed (same hardware) | 6 img/s | 6 img/s | 16 img/s | 2.7× faster |
| Linear probe speed (same hardware) | 2.45 s/task | 2.40 s/task | 0.96 s/task | 2.5× faster |
| Overall performance | All models at least comparable | | | Not generally inferior |
| Classification (Wins at $p < 0.05$, Fig. 2) | 7 wins (2 ties) | 11 wins (7 ties) | 36 wins (9 ties) | >3× the wins |
| Unsupervised projections (Wins at $p < 0.05$, Fig. 3 and 4) | 1 win (1 tie) | 2 wins (3 ties) | 42 wins (2 ties) | >20× the wins |
| External transportability (Wins at $p < 0.05$, Fig. 5) | 6 wins (2 ties) | 17 wins (5 ties) | 41 wins (5 ties) | >2× the wins |

See the following sections for detailed performance on downstream tasks.
See the Methods section for detailed explanations for all of the other rows. A detailed breakdown of statistically significant wins per dataset is provided in Supplementary S9.
*GPU* graphics processing unit, *A100* NVIDIA A100 GPU, *VRAM* video random access memory (GPU memory), *img/s* images per second.

RETFound-MEH. Supplementary S1 provides an overview of the datasets used for training. RETFound-Green's training data is fully open and a strict subset of the training data used for DERETFound. Furthermore, RETFound-Green required two orders of magnitude less computational resources, 400 times less than the original RETFound-MEH and 600 times less than DERETFound which required substantial compute resources for generating synthetic images. Thus, RETFound-Green was trained at a substantially lower cost. This also translates into a smaller estimated carbon footprint[20]: Training RETFound-MEH had an environmental impact comparable to burning 81 kg of coal, DERETFound 117 kg of coal, and RETFound-Green only 0.2 kg of coal.

In addition to being more efficient to train, RETFound-Green is also more efficient in downstream use. The model is 14× smaller and can thus be stored and downloaded more easily, which especially benefits researchers with slow internet connections. Note, that this already factors in an improvement we made to RETFound-MEH and DERETFound that more than halves their file size without loss of performance, as described in detail in the Methods section. It also provides denser embeddings that require 2.6 times less space, which makes maintaining and sharing a vector database of image embeddings more efficient. Obtaining embeddings of images is about 2.6 times faster and thus more accessible even in lower resource settings. Finally, as the embeddings are denser, fitting a predictive model using those embeddings is also faster.

Another harder to quantify yet important practical advantage is ease-of-use and maintainability of the software code. Easier set up and better maintainability is particularly important as users of medical foundation models might be less familiar with programming than those developing them. Specifically, RETFound-MEH requires a five-year-old version of the Python programming language (version 3.7.5) which is now considered "end-of-life" and no longer officially supported [30] which could make it difficult to use in Trusted Research Environments due to security considerations. RETFound-Green works with currently supported versions.

## Performance on diverse downstream tasks
We compare the performance of the three foundation models across six datasets from different continents and a variety of tasks. This includes a Retinopathy of Prematurity (ROP) dataset[30] from China of

young infants, which represents a substantial distribution shift from the pretraining data which only included adults. Second, BRSET[31], a large and richly annotated dataset from Brazil with a variety of downstream tasks. Third, the Indian Diabetic Retinopathy image Dataset (IDRiD), which has high-quality annotations for diabetic retinopathy. Diabetic retinopathy is a leading cause of blindness and projected to affect over 100 million people by 2045[32]. Fourth, we use three datasets that were used in the original RETFound-MEH paper with identical datasplits and pre-processing. This includes two multi-class datasets, namely "Retina" and JSIEC1000, and Messidor2, a diabetic retinopathy dataset from France. Diabetic retinopathy screening and severity grading is a common task in ophthalmology, making it a good test bed for fine-grained disease assessment. We consider diabetic retinopathy detection as well as detection of "referrable" diabetic retinopathy that is more severe than no or mild diabetic retinopathy (grade 2 and higher on the International Clinical Diabetic Retinopathy Scale[33]). All three models have been pre-trained on some data relating to diabetic retinopathy, with RETFound-MEH having the most in both relative and absolute terms. Thus, a priori, we would expect diabetic retinopathy related tasks to favour RETFound-MEH. Finally, we also consider detailed diabetic retinopathy screening and grading in BRSET. We use both the whole dataset as well as subset containing only people with diabetes. The latter mimics typical diabetic retinopathy screening programmes where people are invited due to having diabetes.

The results are shown in Fig. 2. No model outperformed the other models across all tasks. Overall, RETFound-Green performed the best with 36 statistically significant wins compared to 11 for DERETFound and 7 for RETFound-MEH. Interestingly, this also holds true when focusing only on tasks related to diabetic retinopathy, where we might have expected RETFound-MEH to have an advantage due to being pretrained on the most data related to diabetic retinopathy.

### Unsupervised low-dimensional projection of feature vectors
A key aspect of foundation models is the quality of the feature vectors they provide. These vectors should be salient and capture relevant aspects of the input images. In the previous sections, we have examined this by using the vectors for classification tasks. In this section, we

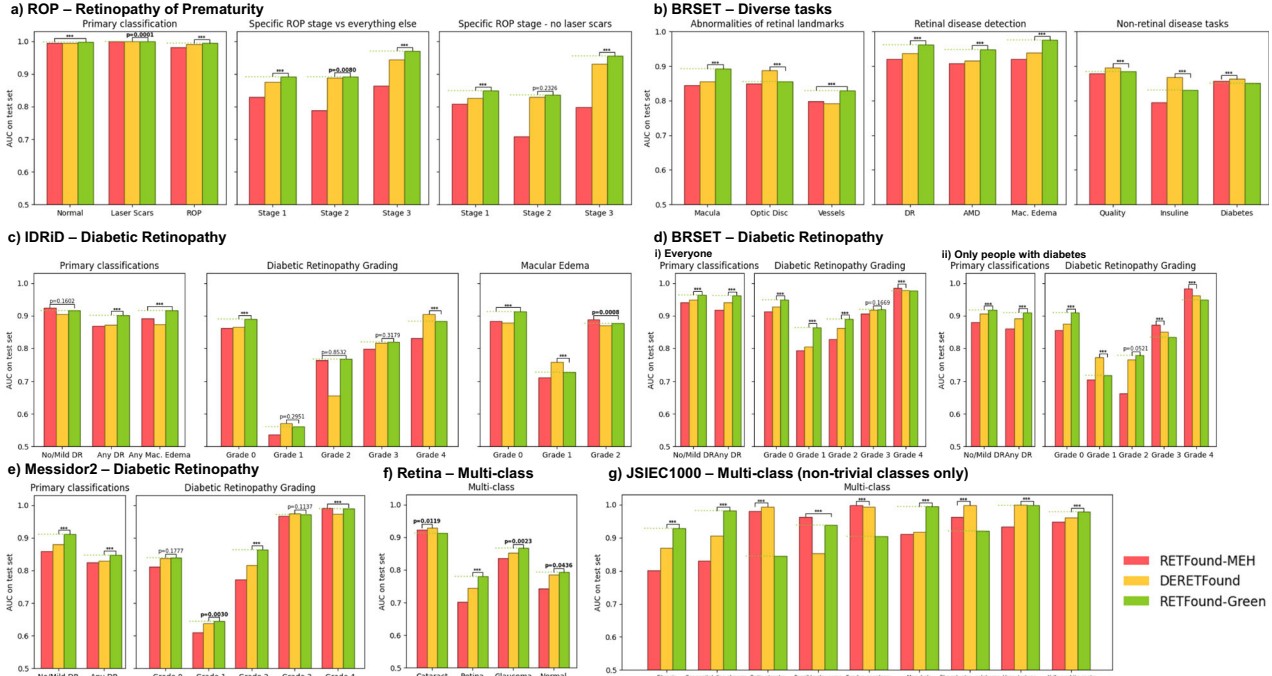

**Fig. 2 | Performance across a variety of models and tasks. A** ROP – Retinopathy of Prematurity, **B** BRSET - Diverse tasks, **c** IDRiD – Diabetic Retinopathy, **d** BRSET – Diabetic Retinopathy, **e** Messidor2 – Diabetic Retinopathy, **f** Retina – Multi-class, **g** JSIEC1000 Multi-class (non-trivial classes only). The horizontal green dashed line indicates the performance of RETFound-Green to aid visual comparison. For robustness, reported results are the median of 100 bootstrap samples of the test set. Best result for each task in bold, the bar with *p*-value indicates the result of a two-sided Wilcoxon signed-rank test between the best and second best methods across the 100 bootstrap samples, with *p* < 0.05 in bold. "***" indicates *p* < 0.0001. AUC Area under the receiver operating characteristic curve, ROP Retinopathy of Prematurity, DR Diabetic Retinopathy, AMD Age-related Macular Degeneration, Mac. Edema Macular Edema, Cogenital disc abnorm. Cogenital disc abnormality, Mac. hole Macular hole, Chor.ret. atro.-coloboma Chorioretinal atrophy-coloboma, Vess. Tortuous. Vessel tortuosity.

look at low-dimensional projections of the feature vectors obtained in an unsupervised way, i.e. without classification labels.

We focus on the ROP and IDRiD datasets here, because there the disease status and severity labels provide a key dimension along which the data should vary. These labels then allow for qualitative evaluation of whether the low-dimensional projection captured this key axis of variation. Additionally, we provide predictive performance (AUC) on the test set using Logistic Regression and k-Nearest-Neighbours (KNN) to objectively quantify whether these two-dimensional projections indeed separate the classes well. We use two popular dimensionality reduction methods: Principal Component Analysis (PCA) and Uniform Manifold Approximation and Projection (UMAP)[34]. Briefly, PCA is a linear method that preserves as much variance as possible, while UMAP is non-linear and aims to represent data points as close to each other in the lower-dimensional projection if they are close to each other in the original high-dimensional space. Detailed explanations can be found in the Methods section.

Figure 3 shows the results for the ROP dataset. For the PCA projections, RETFound-Green has a cluster of "Normal" images on the right, a cluster of "Laser Scars" at the bottom, and a cluster of "ROP" images in the centre and top, with the Laser Scars and ROP images overlapping. More severe ROP appears to be closer to the centre and overlaps more with Laser Scars, whereas less severe ROP is more towards the top of the plot. This matches that laser treatment is done for more severe cases. For RETFound-MEH and DERETFound, the target classes appear to be separated less. RETFound-MEH has a smaller cluster in the top right that does not appear to correspond to ROP status. These visual impressions are supported by the AUCs for RETFound-Green being the highest for all target labels and for both linear Logistic Regression and non-linear, neighbourhood-based KNN. RETFound-Green achieves a statistically significant win for all eleven out of twelve comparisons, with the only exception being "Stage 3

ROP" using KNN, where its advantage over DERETFound is not significant.

For the UMAP projections, all three models appear find three clusters within the data, which do not correspond to ROP status. The ROP dataset was acquired with three different camera systems by different manufacturers, which these clusters could plausibly correspond to, though we did not ascertain this. For all models, the two larger clusters seem to separate between Normal, Laser Scars, and ROP. However, this apparent separation is clearest for RETFound-Green. This is again confirmed by the AUCs, with RETFound-Green achieving a statistically significant win across all twelve comparisons.

For the IDRiD dataset (Fig. 4), we find similar results. In the PCA projections, RETFound-MEH and DERETFound both show two clusters that are each somewhat organised by disease status. RETFound-Green on the other hand has a single cluster that shows a clear transition from Normal to Grade 4. RETFound-Green achieves a statistically significant win for ten of the twelve comparisons, DERETFound has one significant win, and there was one case where DERETFound performed best but the difference with RETFound-Green was not significant. Looking at the UMAP plots, all models, including RETFound-Green, identify two clusters. Unlike the ROP dataset, the IDRiD dataset is described as being acquired with a single type of fundus camera[35] at the same location. However, data acquisition spanned a period of eight years, so some distribution shift may have occurred during that period. In all clusters, Normal images appear to be separated from DR images, with the difference being clearest for RETFound-Green. RETFound-Green has 9 significant wins, RETFound-MEH and DERETFound-MEH have one significant win each, and there was a final case where DERETFound performed best without this being statistically significant.

These results suggest that the feature vectors from all three models capture relevant key axes of variation in the input images, such as disease status, without needing classification labels. The results on

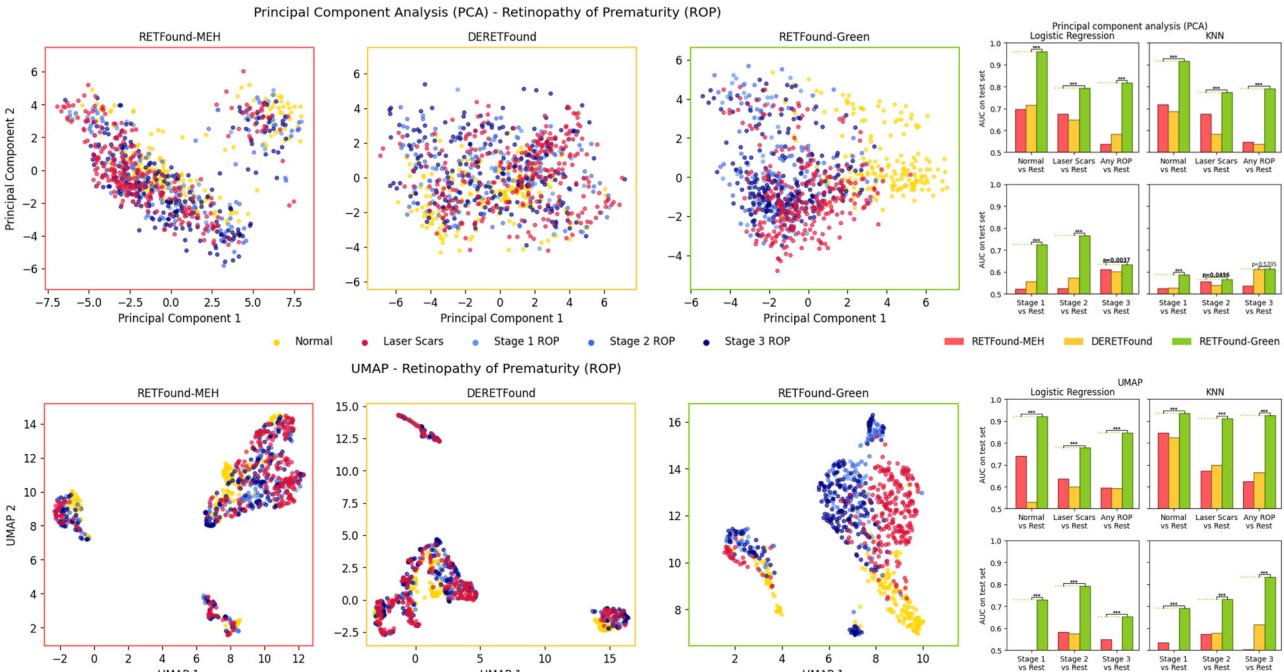

**Fig. 3 | Two-dimensional projections of the features with PCA (top row) and UMAP (bottom row) for the ROP dataset.** Scatter plots show the representations of the train set for each of the three foundation models. The bar plots on the right show the test set AUCs using the two-dimensional projections as input features for different binary targets, using Logistic Regression and KNN as classification algorithms. Missing bars indicate an AUC < 0.5, i.e. worse than random guessing. For robustness, reported results are the median of 100 bootstrap samples of the test set. The horizontal bars indicate the result of a two-sided Wilcoxon signed-rank test between the best and second best methods across the 100 bootstrap samples, with $p < 0.05$ in bold. "***" indicates $p < 0.0001$. AUC Area under the receiver operating characteristic curve, ROP Retinopathy of Prematurity.

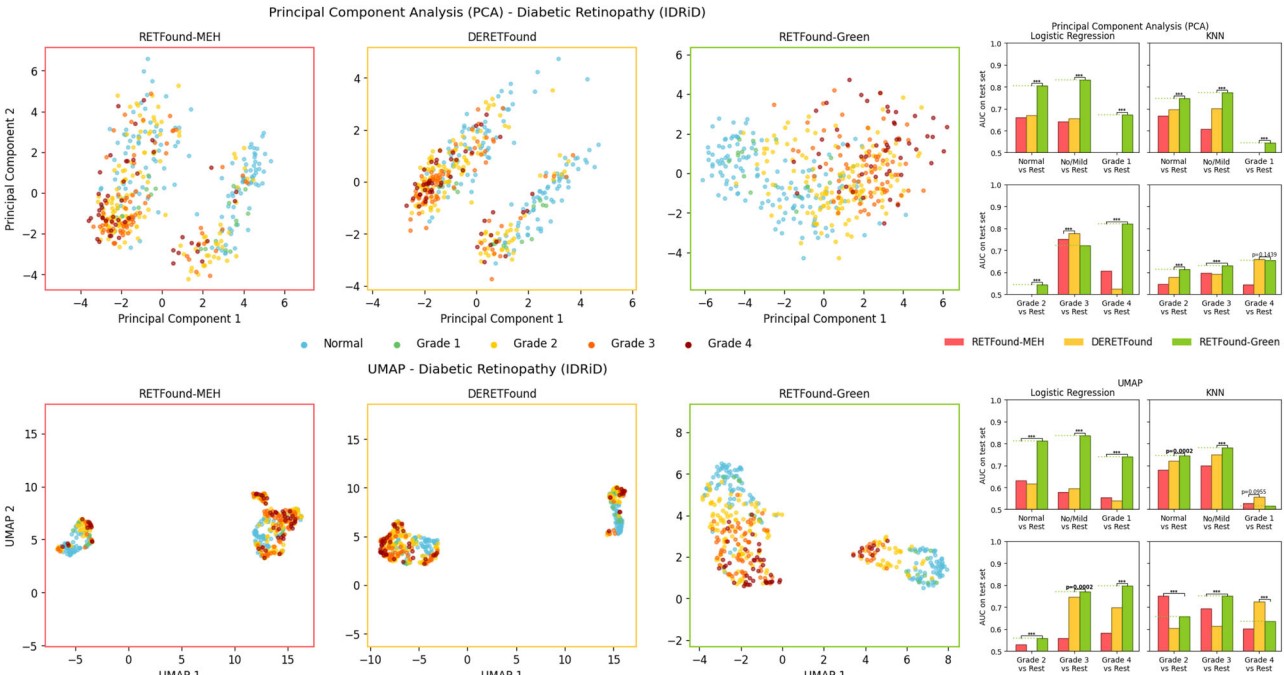

**Fig. 4 | Two-dimensional projections of the features with PCA (top row) and UMAP (bottom row) for the IDRiD dataset.** Scatter plots show the representations of the train set for each of the three foundation models. The bar plots on the right show the test set AUCs using the two-dimensional projections as input features for different binary targets, using Logistic Regression and KNN as classification algorithms. Missing bars indicate an AUC < 0.5, i.e. worse than random guessing. For robustness, reported results are the median of 100 bootstrap samples of the test set. The horizontal bars indicate the result of a two-sided Wilcoxon signed-rank test between the best and second best methods across the 100 bootstrap samples, with $p < 0.05$ in bold. "***" indicates $p < 0.0001$. AUC Area under the receiver operating characteristic curve, DR Diabetic Retinopathy.

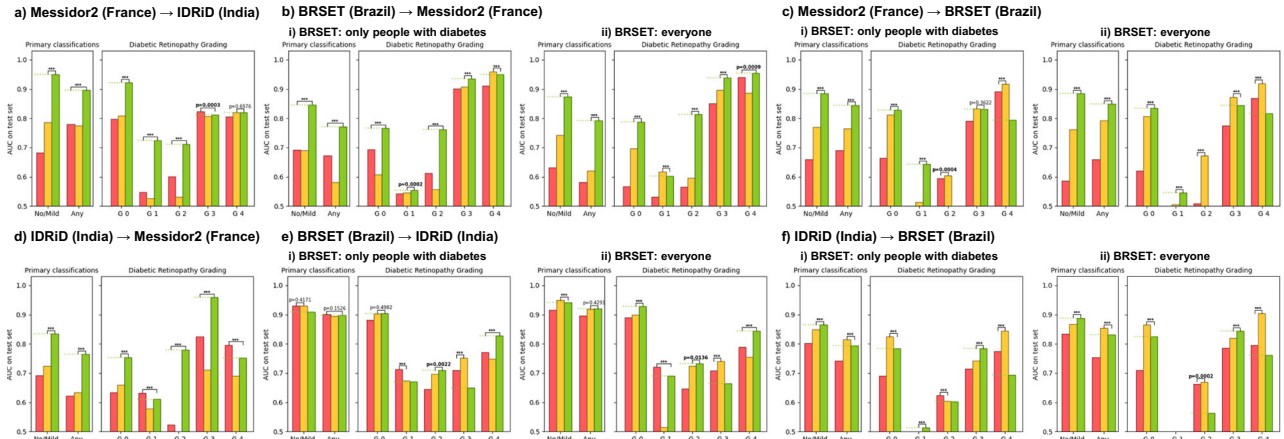

**Fig. 5 | Performance for diabetic-retinopathy-related tasks between IDRiD, Messidor2 and BRSET when training the model on one dataset and evaluating on another. a** Messidor-2 (France) → IDRiD (India). **b** BRSET (Brazil) → Messidor-2 (France): (i) diabetes-only subset; (ii) entire cohort. **c** Messidor-2 (France) → BRSET (Brazil): (i) diabetes-only subset; (ii) entire cohort. **d** IDRiD (India) → Messidor-2 (France). **e** BRSET (Brazil) → IDRiD (India): (i) diabetes-only subset; (ii) entire cohort. **f** IDRiD (India) → BRSET (Brazil): (i) diabetes-only subset; (ii) entire cohort. Missing

bars indicate an AUC < 0.5, i.e. worse than random guessing. For robustness, reported results are the median of 100 bootstrap samples of the test set. The horizontal bars indicate the result of a two-sided Wilcoxon signed-rank test between the best and second best methods across the 100 bootstrap samples, with $p < 0.05$ in bold. "***" indicates $p < 0.0001$. AUC Area under the receiver operating characteristic curve.

the ROP dataset are especially promising as ROP was not present in the pre-training dataset for any of the three models and further comprises of young infants, i.e. a very different demographic compared to the other datasets that comprise of adults.

Visually and quantitatively, RETFound-Green provides the clearest separation of different levels of disease severity, for both PCA and UMAP. This suggests that RETFound-Green provides highly semantically meaningful vector representations, which could make it advantageous not just for lower-dimensional projections of the data but also other tasks where out-of-the-box meaningful feature vectors are needed such as clustering or vector search. We provide additional sensitivity analyses for the UMAP embeddings in Supplementary S2, which provide very similar results to the ones presented here, suggesting that our results are robust.

### External transportability of feature vectors across datasets

Performance within a given setting and population is important, but in medical AI it is crucial that models also perform well in other settings and populations. None of the downstream datasets had been used for pre-training for any of the three foundation models, so the results from the previous two sections already provide evidence that the models generalise beyond their pre-training datasets. However, an additional question that is interesting to consider is whether the classifiers we fit on the feature vectors generalise from one downstream dataset for another downstream dataset. We refer to this as external transportability.

We examine this using the common and well-standardised task of diabetic retinopathy screening and grading as an exemplar. We use the IDRiD, BRSET, and Messidor2 datasets which all use the International Clinical Diabetic Retinopathy Scale[33]. Thus, the definitions for these labels should be consistent, although in practice there will be systematic differences in interpretation between centres[36].

These three datasets have substantial distribution shift between them. First, they are from different healthcare institutions from different countries (India, Brazil, France) on three continents (Asia, South America, Europe). Thus, they capture distinct populations from different healthcare contexts that presumably have very different ethnic makeup, although explicit information is not available. Second, they use different cameras from four different manufacturers: Nikon NF505 and Canon CR-2 for BRSET, Kowa VX-10α for IDRiD, and Topcon TRC

NW6 for Messidor2. Finally, as an additional dimension of distribution shift, in the case of BRSET we also consider two cases: including only patients with diabetes as is the case for IDRiD and Messidor2, and including everyone in BRSET. The former case mimics a typical diabetic retinopathy screening programme where patients with known diabetes are invited, whereas the latter represents a more general screening case, e.g. in settings where the diabetes status of many patients is unknown. Supplementary S10 provides a tabular comparison of these three datasets.

Figure 5 shows the results of these experiments. Overall, all three models generalise to other datasets. RETFound-Green appears to generalise the best, with 41 statistically significant wins compared to 6 for RETFound-MEH and 17 for DERETFound.

Supplementary S3 examines the transportability of the two-dimensional projections obtained with PCA and UMAP between IDRiD and BRSET. We fit the transformations and classifiers to one dataset, and then plotting the projections and evaluate the classifiers on the other dataset. Briefly, we observe that the representations appear to transfer well overall, suggesting that the representations encoded in the vector embeddings meaningfully transfer between different datasets. In Supplementary S3, RETFound-Green appears to provide the clearest separation qualitatively and it achieves statistically significant wins for the majority of the comparisons quantitatively.

These results suggest classifiers fit to the feature vectors obtained from the three foundation models generalise to external datasets, even when they were acquired in a different population. Using the full vectors, all three models showed reasonable performance, with RETFound-Green performing best most of the time. For the unsupervised two-dimensional projections, RETFound-Green showed the best transportability overall both qualitatively and quantitatively. This further supports the findings from the previous section and might suggest that RETFound-Green embeddings could also be used for vector search in heterogeneous databases.

### Fully finetuned RETFound-MEH vs RETFound-Green with simple linear probing

In most of our experiments, we adapt the foundation models using "linear probing", i.e. fitting a small linear model to the vector embeddings but leaving the underlying model unchanged, as opposed to training all of the foundation model parameters. Linear probing has

## a) Messidor2

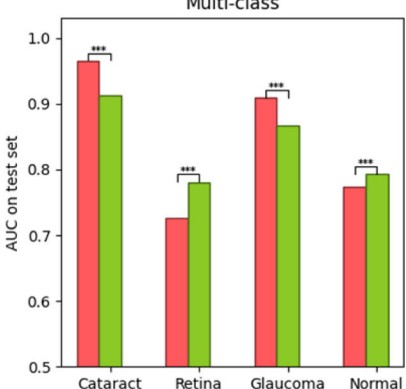

## b) Retina

## c) JSIEC1000 (non-trivial classes)

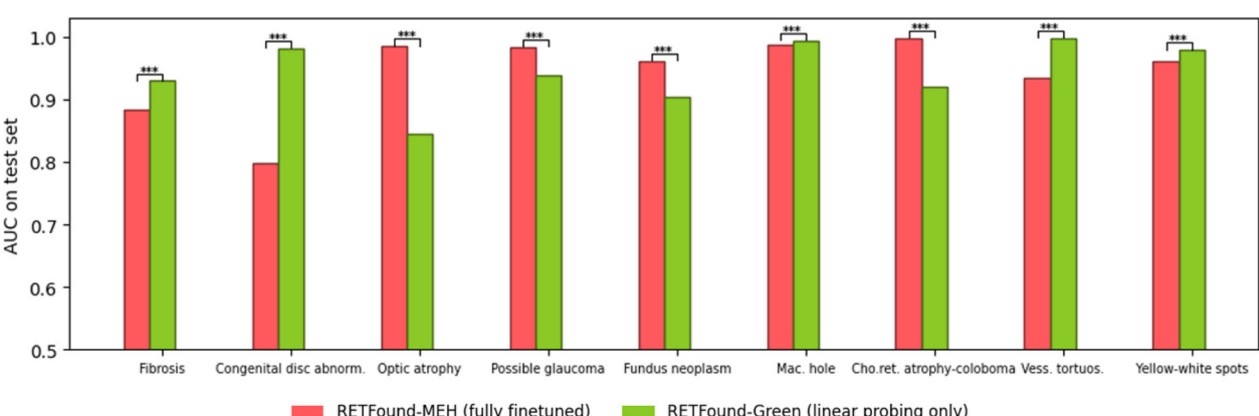

**Fig. 6 | Comparison of fully finetuned RETFound-MEH using the fine-tuned weights released by the original authors with RETFound-Green with only linear probing. a** On the Messidor2 dataset, **b** on the Retina dataset, **c** on the non-trivial classes from the JSIEC1000 dataset. Missing bars indicate an AUC < 0.5, i.e. worse than random guessing. For robustness, reported results are the median of 100 bootstrap samples of the test set. The horizontal bars indicate the result of a two-sided Wilcoxon signed-rank test between the best and second best methods across the 100 bootstrap samples, with $p < 0.05$ in bold. "***" indicates $p < 0.0001$. AUC Area under the receiver operating characteristic curve, DR Diabetic Retinopathy, Cogenital disc abnorm. Cogenital disc abnormality, Mac. hole Macular hole, Chor.ret. atro.-coloboma Chorioretinal atrophy-coloboma, Vess. Tortuous. Vessel tortuosity.

many advantages. First of all, it can be done on low-end hardware like an ordinary laptop whereas full finetuning requires a modern GPU workstation. Second, it can be done from pre-computed vector embeddings without requiring access to the original images. Thus, it would be possible to develop new classifiers using a vector database and in that case fitting the linear probe itself typically takes only a few seconds. Third, each finetuned version of RETFound-MEH needs to be stored separately and yields different vector embeddings. This would make maintaining a vector database impractical, as we would need to store a different vector for each task that the foundation model was finetuned for. Likewise, storing multiple versions of each model is cumbersome. A finetuned version of RETFound-MEH for a binary classification task has 303,302,657 parameters. The linear probe for RETFound-Green only has 385 parameters, one for each dimension of the feature vector and an intercept, almost a million times less.

For these reasons, the linear probing experiments are the most relevant for evaluating the performance of foundation models. Practitioners may occasionally want to finetune the whole foundation model for a specific task, but a good foundation model should deliver versatile, meaningful vector embeddings that can be used for different tasks out of the box. Finetuning also has many more design choices and hyperparameters, which makes a fair comparison more complex than with linear probing.

Nevertheless, it is interesting to consider whether full finetuning offers substantial advantages over simple linear probing. For this purpose, we compare RETFound-MEH with full finetuning and RETFound-Green with only linear probing. We consider the Messidor2, Retina, and JSIEC1000 datasets used in the original RETFound-MEH paper. To provide a maximally fair comparison, we use the exact same data splits and use the finetuned versions of RETFound-MEH that have kindly been released by the original authors. Thus, all design choices and hyperparameters for the finetuning were selected by the RETFound-MEH authors. Additionally, RETFound-MEH split the data into training, validation, and testing sets. The validation set was used to select the best finetuned version for final evaluation. When doing linear probing for RETFound-Green, we do not use the validation set at all. Thus, RETFound-Green effectively has slightly less data available than RETFound-MEH in these experiments.

Figure 6 shows the results of the comparison between fully finetuned RETFound-MEH and RETFound-Green with linear probing only. Overall, both models achieve comparable performance with either model outperforming the other in some comparisons. RETFound-Green has four statistically significant wins compared to three for RETFound-MEH in Messidor2, five wins compared to four in JSIEC1000, and both models have two wins each in the Retina dataset.

These results suggest RETFound-Green can deliver a comparable level of performance with simple linear probing as RETFound-MEH delivers with full finetuning. Given the numerous advantages of linear probing, this is a very strong result. On the other hand, for some tasks RETFound-MEH with finetuning still achieves better performance. This suggests that there is room for improvement yet and future retinal foundation models should aim to unlock this performance with simple linear probing.

## Discussion

This work presents RETFound-Green, a high-performance retinal image foundation model trained using a novel Token Reconstruction pre-training objective. Our approach allowed us to train RETFound-Green with 50% less data and 400× less computational resources compared to the "best of both worlds" of the two previously proposed models, namely DERETFound's data efficiency and RETFound-MEHs compute efficiency. RETFound-Green is also far more efficient and easier to use in downstream applications. Despite these substantial improvements in efficiency, RETFound-Green does not perform systematically worse. On the contrary, in our experiments it achieves twice as many statistically significant wins as the next best model.

While AI-powered advancements in healthcare hold great promise, there is a risk that they could lead to new or exacerbated disparities as the resources necessary to develop or make use of them are not accessible to everyone[37]. We believe that RETFound-Green will not only serve as a useful foundation model for retinal image analysis but also democratize foundation models in multiple ways: First, democratising access to foundation models. RETFound-Green is far more efficient in downstream applications, which will especially benefit researchers with comparatively fewer resources. For example, on a 128 Kb/s internet connection, it would take 2 days and 15 h to download RETFound-MEH or DERETFound as shared by the authors, 19.5 h with our optimisation of their models, but RETFound-Green could be downloaded in just 1.5 h. As RETFound-Green is 2.7 times faster when it comes to calculating embeddings, it could take a workload that previously would have taken a whole day down to less than nine hours, or allow such a workload to be run in the same time on hardware that is 2.7 times slower. If models are used routinely in healthcare, this can produce substantial emissions[19] and increased inference efficiency translates into more sustainable medical AI.

Second, RETFound-Green and the Token Reconstruction objective we introduce democratise foundation model development by requiring far less data than before: half of the data-efficient DERETFound and 12 times less than the original RETFound-MEH. This not only allows researchers without access to massive databases to develop their own foundation models, but could also enable future foundation models to be substantially fairer than current ones. As the required dataset size is now less than 100,000 images, it would be feasible to collect a comparatively small yet representative and unbiased dataset that includes data from diverse patients in terms of ethnicity, sex, disease status, etc. and from a variety of healthcare contexts across the world.

Third, the reduced computational cost of our pre-training objective democratises foundational models by making it accessible to researchers without substantial financial and computational resources, as well as by making it more sustainable by reducing the environmental impact which makes it fairer towards future generations and current generations in developing countries, who are most affected by the impact of climate change. Concretely, the previous approaches required over a hundred days of high-end datacentre GPU days that would cost thousands of dollars, whereas our model was trained for 8 h on a consumer-grade GPU that could be found in a high-end "gaming computer". By increasing the training time to a day or two, it would be possible to adapt our approach to instead also run on a modern low-end gaming computer, which would be within reach of many

researchers globally. The massively reduced environmental impact means that future scaled-up foundation models can be developed in good conscience.

It is important to note that our approach, including the Token Reconstruction objective, are in no way specific to retinal imaging and likely would work similarly well for developing foundation models for other imaging modalities, both medical and non-medical. We focus on retinal imaging is simply because we are familiar with this domain. Given the order-of-magnitude gains in efficiency we observe, we think it is imperative that we share our results promptly to hopefully avoid new foundation models being trained with less efficient methods which would be highly wasteful based on our results. Furthermore, RETFound-Green already offers many practical advantages for researchers working on retinal image analysis. Thus, we chose not to delay disseminating our results, instead of demonstrating effectiveness across a range of modalities first.

While our Token Reconstruction objective appears to be very effective, we do not claim that this is already the best possible strategy for developing foundation models. In fact, we did not tune or iterate on our approach at all, thus it would be surprising if there was no room left for improvement. However, at present our approach is already vastly more efficient than the approach used by RETFound-MEH and DERETFound. One important general lesson from this is that when developing models for specific domains, it is likely very suboptimal to simply copy the approach taken by researchers in general, natural image computer vision. The approaches are developed for datasets that are orders of magnitude larger and to train models entirely "from scratch", i.e. using randomly initialised parameters.

RETFound-Green defies conventional wisdom that large amounts of data and compute are necessary for the current levels of performance by achieving the same level of performance with much less resources. This raises the possibility that a better version of RETFound-Green could be trained with a similar amount of resources to what was used for RETFound-MEH and DERETFound.

RETFound-Green is not only more efficient to train, but also more efficient in downstream usage. In principle existing foundation models, such as RETFound-MEH and DERETFound, could be made more efficient for downstream usage through pruning of less important model parameters or by distilling them into smaller models. However, such approaches would not address the resources required for the initial training. Indeed, they would require additional computational resources and tend to lead to reduced model performance. RETFound-Green is more efficient for downstream usage out of the box while being high-performance, too.

There are two additional potential benefits of RETFound-Green that we do not investigate in the current manuscript but that are important to note. First, the smaller model size would be a substantial benefit for federated learning, where the data remains distributed across multiple sites (e.g. different hospitals) and not centrally aggregated. A key bottleneck for federated learning is that either model parameters or gradient updates need to be sent back and forth at each iteration, many thousands or even hundreds of thousand times. For RETFound-MEH and DERETFound, this would require over 1GB of data transfer for each iteration, whereas for RETFound-Green it would only be 0.09GB, which would result in massively reduced training time and costs. Second, RETFound-Green might be more privacy-preserving. The MAE pre-training objective involves reconstructing the exact pixels of the input image, whereas our Token Reconstruction objective focuses only on more abstract features. The RETFound-Green embeddings are also denser – 384 vs 1024 numbers per image – which substantially reduces the risk of the embeddings allowing specific patients to be identified or images to be reconstructed.

Our results are strong overall, but there are a number of limitations for this work. First, our downstream evaluations focused on six

datasets from different countries, continents, and healthcare contexts but our experiments are not exhaustive. Notably, none of the datasets we used are from Africa. While we compare the models across a variety of tasks and find that RETFound-Green generally performs best, it is possible that for other datasets and tasks the relative performance of the three evaluated models could be different. At present, the most important insight is that RETFound-Green does not perform systematically worse while being far more efficient, which is well supported by the data. However, RETFound-Green might represent an improvement in performance and especially the results for the unsupervised low-dimensional projections are very encouraging. Future work should compare the models across more datasets and an even broader selection of tasks. Specifically, this should include classification of disease severity and subtype rather than simple binary tasks as well as progression prediction. Second, although achieving the same level of performance with far fewer resources suggests that even better performance could be achieved when scaling our approach up, we do not attempt this in the current manuscript as we do not have access to that level of resources ourselves. This is a very interesting possibility that should be explored by future work. Third, while our approach is not specific to retinal images and likely could be applied to many different imaging domains, we likewise do not investigate this in the current manuscript. Fourth, future work should develop our Token Reconstruction method further and investigate the impact of design choices such as model architecture, original pre-training method, or parameter count. Fifth, our approach being generic rather than specific to retinal images is also a weakness as adding elements specific to retinal image analysis and ophthalmology, such as integrating additional information about the patient and their symptoms or considering longitudinal images, could further improve the utility of our model for practical applications, which we plan to investigate in future work. Finally, while foundation models are a tool that may accelerate and improve AI model development, when developing a model for clinical deployment, all the same considerations as before apply. This includes carefully considering the quality and relevance of the training data used to adapt the foundation model, conducting careful evaluation that matches the intended usage, training and evaluating on realistic image quality rather than just pristine images, while rejecting unanalysable images automatically when deployed, and considering how the AI model's outputs will fit into clinical workflows (e.g. should a diabetic retinopathy model also flag up other pathologies?).

In conclusion, we present RETFound-Green trained with a novel Token Reconstruction approach that allows us to match and possibly exceed the performance of previous retinal image foundation models with far less resources while being substantially more efficient and easier to use in practice. This is useful for researchers in the field and could democratise both access to and development of foundation models in retinal imaging and beyond.

## Methods

### RETFound-Green

Like RETFound-MEH and DERETFound, RETFound-Green uses the original vision transformer architecture[27] but with four extra "register tokens"[38] that do not relate to a specific patch. This is a minor and straightforward modification of the architecture that has been observed to yield smoother attention maps by providing a register for the model to capture additional global context. They add negligible computational cost and reconstructing them might help our model to better capture global context. More recent and advanced vision transformer architectures could be explored in the future, for example using group tokens[39] or query pooling[40] to reduce the internal dimensionality. In this work, we chose a similar architecture to make the comparison more direct. RETFound-MEH and DERETFound use the "large" version of the vision transformer, RETFound-Green "small" version. There is also a "base" version, so the small one is two steps

down and much smaller. This allows RETFound-Green to process the images at higher resolution of $392 \times 392$ instead of $224 \times 224$ used by the other two models while still being more efficient at the same time. However, the results for RETFound-Green trained at $224 \times 224$ (Supplementary Fig. 6) show that our approach is effective even at the same resolution.

RETFound-MEH and DERETFound start with pre-trained weights from the Masked Autoencoder (MAE) paper[22] and then train these weights on retinal images using the MAE strategy and codebase. We start with weights from DinoV2[41], a self-supervised learning method like MAE. However, unlike the other two foundation models, we only use the DinoV2 weights and then train those on retinal images using our own Token Reconstruction strategy, described in the next section, and our own codebase. The DinoV2 self-supervised learning approach or codebase are not used.

### Self-supervised training via token reconstruction

We propose a novel method for self-supervised training of pre-trained models. We note that MAE[22] and DinoV2[41] were devised for training models entirely from scratch, i.e. using randomly initialised weights. These strategies are very computationally expensive and require vast amounts of data. When developing foundation models for medical imaging, we can start with models that were already pre-trained on natural images. In fact, this is the same approach as RETFound-MEH and DERETFound use. However, they not only use the pre-trained MAE weights as a starting point but the MAE self-supervised learning objective, including the hyperparameters. This might not be optimal as the MAE strategy and the hyperparameters were chosen for training from scratch on natural images, which is complex and resource-intensive, especially for vision transformers.

When adapting the existing model to retinal imaging, we have two objectives for our self-supervised training approach. First, instilling knowledge about the general appearance and structure of retinal images. Second, optimising the features of our model for this space without unlearning existing, useful features. We propose a novel token reconstruction strategy that is straight-forward yet effective: We take a frozen version of the model we are adapting, pass retinal images through it and obtain the output tokens. Our model, a second non-frozen copy, is given noisy versions of the same images as input and its output tokens are compared to the first model's output tokens, and our model incurs a loss for how different the two are. In other words, our model has to reconstruct the output tokens from noisy inputs. Our approach is illustrated in Fig. 7.

The intuition behind the Token Reconstruction objective is that to match the output of the frozen model, our model needs to use information from the non-corrupted input tokens to predict the correct outputs for all tokens. This requires understanding the structure and co-dependencies of retinal images. For instance, if a corrupted patch contains lesions, then it is likely that the rest of the image either also contains lesions or other anatomical features that make the presence of lesions more probable. Furthermore, some anatomical structures occur in specific locations such as the optic disc or macula. To reconstruct these, the model benefits from learning what structures tend to occur in which locations. Likewise, the pixel-wise noise needs to be ignored by our model, which requires understanding what is part of the original image and what is added noise.

We use two simple noise strategies: First, a random subset of input tokens is replaced with a trainable "corruption token" which effectively erases the information about a patch of the input image. Second, we apply pixel-wise Gaussian noise to the input image. The detailed parameter settings are described in the section on training parameters below. Our strategy is straight-forward to implement and computationally efficient. We note that there are many possible extensions of approach that could be tried, for example additional strategies of

**A)** Masked Autoencoder pre-training strategy

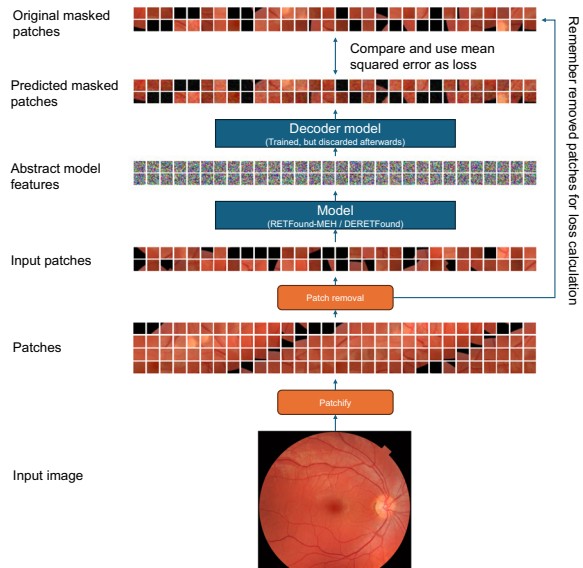

**B)** Token reconstruction pre-training strategy (ours)

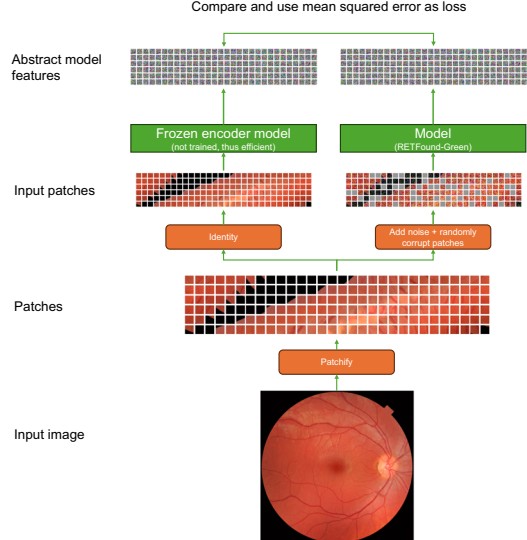

**Fig. 7 | Comparison between the Masked Autoencoder (MAE) strategy. A** Used by RETFound-MEH and DERETFound with our proposed Token Reconstruction strategy (**B**) that we used to train RETFound-Green. Our strategy focuses on the abstract features that are used in downstream applications, whereas MAE focuses on exact pixel values. MAE requires training a decoder model which increases computational cost but this model is discarded after training. Our strategy also uses an auxiliary model, but this is kept frozen and thus gradients do not need to be computed which lowers computational cost substantially.

making the image noisy, or reconstructing the outputs of multiple different frozen models at the same time.

Though we do not investigate it in detail in this work, we think that the fact that our method reconstructs patches in a perceptual, abstract space as opposed to pixel space is a key advantage. MAE models trained to minimise mean squared error tend to produce blurry reconstructions, as can be seen in both the MAE and RETFound-MEH/ DERETFound papers. This is because the optimal solution for minimising mean squared error is to predict the conditional expectation of the masked patches $y$ given the non-masked patches $x$, $E[y|x]$. Intuitively, this expectation is a weighted average of all possible reconstructions of the partially masked image, each possible reconstruction weighted by how likely it is. Averaging over all possible reconstructions produces a blurry image. The official MAE GitHub repository provides a model trained without their pixel normalisation, which yields better-looking visualisations but worse representations. They even provide an additional model trained with an adversarial loss to produce more realistic and sharper images (https://github.com/facebookresearch/mae?tab=readme-ov-file#visualization-demo) for visualisation purposes, with the adversarial loss forcing the reconstruction towards a particular possible reconstruction. While not discussed in the paper, presumably the adversarial loss also comes at the cost of yielding worse representations for downstream adaptations of the model.

With our Token Reconstruction approach in the other hand, the prediction target is a feature vector providing an abstract description of the input patch. Return to our example above of a masked patch being highly likely to contain small diabetic lesions. Their exact shape and location will be hard to predict, and averaging over all possible reconstructed patches with lesions as MAE encourages will yield a uniform patch with the background retina hue. Thus, encoding the presence of lesions would not substantially improve the loss over just encoding the background hue. Whereas for our Token Reconstruction objective, the optimal prediction will be a weighted average of all abstract representations of the masked patch. We conjecture that the average over all abstract representations of many possible patches

with small lesions will be a vector that still captures the presence of such patterns.

For natural image datasets like ImageNet, there is a lot of high-level variation in the images, which in the machine learning community are referred to as "low-frequency patterns". Consider two images of cars. The cars might not only be different makes and colours, but the images might be taken in different lighting conditions, from different angles, with different amounts and kinds of backgrounds. Now consider that ImageNet contains very diverse classes of objects, from vehicles to specific dog breeds to specific foods. Even a blurry reconstruction of such diverse images requires that the MAE encoder captures key details relevant to the class of the input image. In retinal imaging, on the other hand, all images are comparatively highly similar and very small structures such as a few small lesions can be key to differentiating a healthy retina from an unhealthy one. Thus, we think that doing the reconstructions in an abstract space is a key advantage in retinal imaging. Future work should investigate this in more detail, for example by adapting the different retinal foundation models for segmentation.

### Datasets for pre-training
We used three publicly available datasets for pre-training: AIROGS[42], DDR[43], and ODIR- 2019. We used all images in DDR and ODIR-2019 and then selected a random sample of 53,327 from AIROGS to achieve our desired target amount of exactly 75,000 images. We make the subset of images used available on our GitHub to aid reproducibility. DERETFound used the same three datasets for pretraining plus an additional dataset and slightly over twice as many images as our model. For a more detailed comparison of the pre-training datasets for the three models, please refer to Supplementary S1. None of these pre-training datasets were used in any of the downstream evaluations.

### Datasets for downstream evaluations
For our downstream evaluations, we used the Brazilian Multilabel Ophthalmological Dataset ("BRSET")[31] which contains very rich annotations that enable us to compare models across a variety of tasks. We

randomly split BRSET into training (80%) and testing (20%) sets at the patient level, ensuring that no patient is present in both sets to avoid data leakage. These splits are also made available on our GitHub. We also used the Retinopathy of Prematurity (ROP) dataset[30] from China which we split with the same procedure as for BRSET. The ROP dataset contains images from premature infants. Additionally, we used the Indian Diabetic Retinopathy Image Dataset ("IDRiD")[35] where we use the official train-test-split. Finally, we use the Messidor2[44,45], Retina[5] and Joint Shantou International Eye Center ("JSIEC1000")[46] datasets that were used as downstream tasks by the original RETFound-MEH authors.

The JSIEC1000 dataset has 39 classes in total. However, many of these classes are trivial in that all models achieve a perfect or near-perfect AUC. We focus on the non-trivial classes which we define as those where at least one of the three models we compare achieves an AUC less than 0.95. These are 9 classes, namely 'Fibrosis', 'Congenital disc abnormality', 'Optic atrophy', 'Possible glaucoma', 'Fundus neoplasm', 'MH' (Macular hole), 'Chorioretinal atrophy-coloboma', 'Vessel tortuosity' and 'Yellow-white spots-flecks'. Note that we still retain the other classes as negatives to avoid making the dataset easier by reducing the diversity of other conditions. We use the same 9 classes when comparing the fully finetuned RETFound-MEH with RETFound-Green using only linear probing. For completeness, the full results on all classes are reported in Supplementary S7.

All evaluations use the exact same data splits for all models. For the avoidance of doubt, no part of these downstream datasets were used in the pre-training of any of the three foundation models we compare. Supplementary S6 provides additional information about the downstream datasets.

Before applying our data pipeline during training, we resize all images to a resolution of 1024 × 1024 and save them with very high JPG quality. Colour fundus images are circular and thus the images should be square. However, in some datasets there is considerable horizontal black space. Naïve resizing to a square resolution, as is commonly done in the literature, does not preserve the aspect ratio and leads to the images being squashed. Some images are cropped slightly at the top and bottom, which also gives them a non-square aspect ratio. We detect the fundus area and then crop horizontal black space, or pad the top and bottom, to ensure the images are properly cropped and square before resizing.

We implement this purely because we think it is a more principled way to resize retinal images in a way that preserves the aspect ratio. In principle, this resizing approach could offer an advantage in downstream performance. In our opinion, this is unlikely, especially given that we conducted our downstream evaluations without using this pre-processing approach.

For downstream evaluations, we simply resize the images to the desired size and normalise by each model's specific normalisation constants. Thus, we use the same preprocessing for all three models during the downstream evaluations. This matches how the images were resized for RETFound-MEH and DERETFound to ensure a fair comparison. We intentionally do not use our more complex preprocessing on the downstream tasks to make the comparisons fair by ensuring that better preprocessing plays no role. For the Messidor2, Retina and JSIEC1000 datasets, we use the preprocessed versions provided by the RETFound-MEH authors.

### RETFound-Green training parameters

RETFound-Green was trained for 120 epochs, batch size of 128, using AdamW[47] with a maximum learning rate of $5*10^{-5}$, betas $\beta_1 = 0.9$, $\beta_2 = 0.99$ and weight decay $5*10^{-4}$. No weight decay is applied to the bias parameters, a common practice in modern deep learning. We use a cosine learning rate scheduler[48] with a warmup of 10 epochs where the learning rate linearly increases from its minimum $5*10^{-9}$ to its maximum and a cool-down of 20 epochs where it is kept at its minimum.

We use automatic mixed precision with bfloat16, a "half-precision" data type that uses 16 instead of standard 32-bit floating point numbers that is optimised for deep learning. This reduces memory consumption and speeds up training. The maximum gradient norm is clipped to 0.1.

For the Token Reconstruction objective, we use a corruption ratio sampled from $U(0, \frac{1}{3})$, a pixel corruption with noise sampled from $N(0, 0.2)$. The last image in each batch is kept uncorrupted so the model is robust to the absence of corruption tokens. As loss function we use simple mean squared error. For the projection, we use a small residual MLP consisting of LayerNorm, a linear layer, a GELU activation, and another linear layer. Each of the linear layers had a dimension of 384, same as the dimensionality of our model. The projector takes the model's final representation as input, and the projector's output is then scaled by an element-wise learnable parameter and added to the final representations before loss computation.

We apply slight colour jitter and rotations 25% of the time, pad the sides of the images by random value between 33 and 150 pixels to simulate poorly cropped images 10% of the time, and scale the image with a random value between 80 and 120%. We then use one of three ways to obtain an image of our target resolution of 392 × 392: standard resizing, or a random resized crop with a scale of 70–100%, or a centre crop after scaling the image up by 30%. Note that first properly cropping the images and then simulating poor cropping randomly, we decorrelate poor cropping from specific datasets, which might make our model more robust.

RETFound-Green uses simple normalisation with mean and standard deviation parameters of 0.5 for all three colour channels, whereas RETFound-MEH and DERETFound use the statistics of the ImageNet dataset of means 0.485, 0.456, 0.406 for red, green, and blue channels, and standard deviations of 0.229, 0.224, 0.225. The model learns to adjust to these constants during training, so this is a very small optimisation to increase convenience in downstream use as users of RETFound-Green only need to remember 0.5, instead of looking up these values.

We use identical settings for the version at a lower resolution of 224 × 224, noting that efficiency likely could be improved by increasing the batch size and scaling down the learning rate accordingly. However, we did not tune these parameters for RETFound-Green nor the lower resolution version. Instead, we chose reasonable values that worked well out of the box.

### Computational resources

We measure compute in terms of "A100 days", which means the equivalent of training on a single Nvidia A100 Tensor Core datacentre GPU for 1 day. The original RETFound was trained on 8 A100s for two weeks, so 8*14 = 112 A100 days. DERETFound used 2 A100 days for training the diffusion model, about 113 days for generating images with that model, and finally 8 A100s for 6 days for training the DERETFound model itself, so 2 + 113 + (6*8) = 163 A100 days. RETFound-Green was trained for 8 h on a desktop computer with a single Nvidia RTX 4090 GPU, a consumer-grade "gaming" GPU card. A 4090 is roughly equivalent to 0.82 the performance of the slowest A100, the 40GB cloud version, in the lambda labs benchmarks[49], or 0.63 compared the faster A100 80GB. We take the multiplier that is least favourable to RETFound-Green and estimate that it used (8/24)*0.82 = 0.27 A100 days. Our approach is much more compute efficient during training as we train a smaller model without an additional decoder for fewer image-iterations, yet our Token Reconstruction objective allows us to achieve high performance nonetheless.

For the estimated training costs, we use the on-demand price for a 8x A100 40GB cloud machine on Amazon Web Services of $32.77 per hour (https://aws.amazon.com/ec2/instance-types/p4/) and round it down to $30. Prices differ across providers and regions, and change over time, so these are ballpark estimates. DERETFound uses the 80GB variants for pre-training which are more expensive but we use the

same, lower price of the 40GB variant for all methods. We take the price per day and multiply it by the estimated A100 days divided by 8, as each machine has 8x A100. This gives $10,080 for RETFound, $14,670 for DERETFound, and $24.30 for RETFound-Green. We round the other two methods down and RETFound-Green up to "<$100".

For estimating the CO2 released, we use the Machine Learning Emissions Calculator (https://mlco2.github.io/impact/) proposed by previous work[20]. For a fair comparison, we use the same cloud provider and region, Amazon Web Services in US West (N. California), a rough mid-point between the UK and China, the two countries the compared foundation models were developed in. We think that this is fairer than taking into account the local energy mix, as we want to compare methods and not favour researchers based on where they happen to be located. For example, in Scotland where we are based, renewables provided 113% of the electricity consumption in January 2024[50] and our model was trained overnight, when energy consumption tends to be low. Thus, we likely did not lead to any CO2 being released, but this is just due to our location, not due to our methods. Thus, using the calculator and same cloud provider and region, we estimate RETFound-MEH to have generated 161 kg CO2 equivalent or 81 kg of coal burned, DERETFound 234 kg CO2 equivalent or 117 kg of coal burned, and RETFound-Green 0.39 kg CO2 equivalent or 0.2 kg of coal burned.

The speed at which the vector embeddings are computed is estimated on the same hardware, a low-end workstation with a 12$^{th}$ Gen Intel i5-12600k CPU and an Nvidia RTX3060ti GPU, a last generation, low-end gaming card. This level of hardware is very accessible, even in comparatively low resource settings. For equal comparison, we measure inference speed using GPU-acceleration with full float32 precision and a batch size of 1, following the example script released by Zhou et al. (https://github.com/rmaphoh/RETFound_MAE/blob/main/RETFound_Feature.ipynb). Batching and mixed precision could further improve inference time. The same CPU is used for fitting of the linear probes.

### Storage space for model weights and vector embeddings

Users of foundation models incur storage costs for the model itself, and additionally for each vector embedding. For the model weights for RETFound and DERETFound take up 3.68GB each as shared by their original authors. However, we observe that this includes the image decoder and optimizer states, both of which are only used during training and not necessary in downstream use. The optimizer states are especially expensive as they contain one value for each parameter of the foundation model. By removing both of those and only leaving the weights of the foundation models themselves, we can optimise their file size to 1.12GB, a reduction of 58%. We use this optimised version for comparison to be maximally fair. RETFound-Green takes up 83.8MB, which we round up to 0.09GB.

RETFound and DERETFound have a vector embedding feature dimension of 1024, RETFound-Green's dimension is only 384 and thus its vector embeddings require 2.67× less storage space. Storing 1 million embeddings would take about 39.1GB for RETFound and DERETFound, and 14.6GB for RETFound-Green without any compression, storing standard 32-bit floats.

### Predictive modelling for downstream tasks

We obtain vector embeddings for the images from each of the foundation model and then fit a linear model with logistic linkage function to the downstream training set, also known as "linear probing" in the machine learning community. We use default values in the popular scikit-learn[51] machine learning library, except for increasing the maximum fitting iterations to 20,000 to ensure that all models converge successfully. Note that in machine learning, unlike traditional statistics, a L2-weight penalty is used by default which substantially improves convergence and tends to increase predictive performance. We

standardise the data to 0 mean and unit variance using scikit-learn's StandardScaler to help convergence and ensure the weight penalty has the same effect for all variables. We follow machine learning best practices and estimate the parameters for standardisation only on the training set, which avoids data leakage and simulates the scenario where the test data is not available at training time. For the five-way diabetic retinopathy grading (grades 0–4) and three-way macular edema (grades 0–2) tasks, we fit a single multinomial logistic regression which is the recommended default for scikit-learn. Otherwise, a binary model is fitted per target.

This strategy for adapting foundation models has many advantages over fine-tuning the whole model. First, it can be done using only vector embeddings and labels without requiring the images themselves. This allows developing new models just from vector databases. Second, it is very computationally efficient and can be done on low-end hardware, especially if the vector embeddings are already computed. But even if not, this strategy only requires inferences on each image once (also known as a "forward pass" in machine learning), whereas fine-tuning typically requires multiple passes over each image (once per epoch) and each of those passes is computationally more expensive as in addition to the forward pass we also need to compute gradients and take optimisation steps (also known as "backward pass"). Third, it requires much less technical know-how than fine-tuning a deep learning model and could be done in statistical programming languages like R that clinicians are more likely to be familiar with as well as Python, whereas deep learning is best done in languages like Python or C++.

### Lower dimensional projections using Principal Component Analysis and UMAP

Principal Component Analysis (PCA) is a widely used standard method for dimensionality reduction. PCA linearly decomposes the original high-dimensional dataset into "principal components" which are orthogonal to each other (i.e. they are uncorrelated) and explain a maximum amount of variance. One interpretation of PCA is that it centres the data around the origin, rotates it so that the axes are aligned with the directions of the most variation, and then scales each axis according to how much variance it explains. Another interpretation of PCA is as a linear autoencoder: if one wanted to linearly compress a d-dimensional dataset to k < d dimensions while preserving the maximum amount of variance, then the first k principal components are the optimal solution. Uniform Manifold Approximation and Projection (UMAP)[34] on the other hand is a non-linear method that solves an optimisation problem to reduce a d-dimensional dataset to k < d dimensions such that data points that neighbour each other in the original d-dimensional space also neighbour each other in the k-dimensional space. In other words, UMAP preserves the local structure of the data. However, UMAP aims to additionally keep the distance between different clusters of points meaningful.

We use these two well-established dimensionality reduction methods – linear PCA, non-linear UMAP – to provide an unbiased picture of how well the feature vectors of the compared foundation models capture key axes of variation in the data in an unsupervised way, which is a proxy for evaluating how semantically meaningful these vectors are.

We use scikit-learn's PCA implementation and the umap-learn Python package. For UMAP, we set the random_state and disable multi-threading to make the projections deterministic. As is the case for the predictive modelling, we keep hyperparameters to their default values with the exception of setting the distance metric of UMAP to "cosine" as opposed to the default "Euclidean". The cosine distance between two feature vectors a, b is their dot product normalised by the product of their magnitudes $cosine\ distance(a, b) = \frac{a \cdot b}{\|a\| \|b\|}$. Intuitively, cosine distance measures the angle between two vectors, i.e. whether they are pointing in the same direction from the origin or not, and is insensitive

to their magnitude. Cosine distance is generally preferable in the case of high-dimensional vectors and especially feature vectors extracted from neural networks. However, as a sensitivity analysis we also consider UMAP with Euclidean distance. Furthermore, we additionally consider UMAP with cosine and Euclidean distance applied to the first 100 principal components to ensure that the different dimensionality of the feature vectors does not play a role. These sensitivity analyses can be found in Supplementary S2 and provide qualitatively similar results, suggesting that our results are robust. Given that the distances in the 2-dimensional space should be meaningful and that convergence is not an issue in two dimensions, we do not additionally scale the data when fitting classification models to the 2-dimensional projections.

Since we do not use classification labels to obtain the unsupervised 2-dimensional projections, we can plot the training set data points and colour them by their label to qualitatively assess whether the feature vectors of a given foundation model provided separation between the different classes. To augment this qualitative impression, we additionally provide classification performance obtained by fitting a classifier to the 2-dimensional projection of the training set and evaluated on the test set. We use logistic regression as well as k-Nearest-Neighbours (KNN). Logistic regression in two dimensions finds a single direction that corresponds to increasing likelihood of belonging to the positive class. KNN classifies a point by predicting the probability of a query point belonging to the positive class by averaging the class labels of the k-nearest training points. Thus, KNN can deal with multiple clusters unlike logistic regression. We use scikit-learn for KNN and keep the default setting of k = 5. For better direct comparability to KNN, we use a single binary logistic regression model for each target. Since convergence is not an issue in two dimensions and the magnitudes of the projections should be meaningful, we do not rescale the data for logistic regression. Finally, we again follow best practices and fit the PCA/UMAP projection to the train set only and then apply the same projection to the test set.

### External transportability of classifiers and lower-dimensional projections for diabetic retinopathy

For the experiments looking at external transportability between BRSET, IDRiD and Messidor2, we adopt the same set up as in the previous experiments. For the classification experiments, the logistic regression is fitted to the feature vectors and labels for one dataset and then applied to the other dataset. We use the International Clinical Diabetic Retinopathy Scale grades in both datasets[33]. Similarly, for the lower-dimensional projection experiments, we fit the transformation to one dataset and then transfer it to the other. For the classification results associated with the lower-dimensional projections, we transfer both the transformation and classifier. For ease of implementation, we use the IDRiD training set when fitting to IDRiD and the test set when transferring to IDRiD, while we use all of BRSET and Messidor2 for training or testing, respectively. Since we fit new classifiers for this section specifically, there is no risk of data leakage. The classifier fitted to a given dataset is only evaluated on another dataset.

### Evaluation metrics and statistical analysis

We focus on the Area Under the Receiver Operator Characteristic curve (AU-ROC, often just AUC for short), a widely used ranking metric that summarises sensitivity and specificity across all possible decision thresholds. That makes the AU-ROC preferable for general comparisons over metrics that require binarization of predictions as it captures more information.

A common claim in the literature is that for datasets with class imbalance, the Area Under the Precision Recall curve (AU-PR) should be preferred. However, this not well supported by evidence, as discussed in recent work by McDermott et al.[52] who further show

theoretically and empirically that AU-ROC is robust to class imbalance, while selecting models using AU-PR can lead to disparities across subpopulations. In their conclusion they explicitly state that the AU-ROC might be desirable in domains like healthcare, while AU-PR might not be reliable "in settings where equity and fairness are imperative". Thus, we focus on the AU-ROC in this work.

To ensure robustness of metrics and calculate $p$-values for model comparisons, we first bootstrap the test set 100 times, compute the AUC for each sample, and then report the median value. This captures uncertainty over different test set distributions and ensure that our metrics are not unduly influenced a few individual samples. We note that logistic regression with weight penalty has a unique solution up to tiny differences due to floating point precision and is thus deterministic, so there is no uncertainty in the model fitting procedure.

We then do a two-sided Wilcoxon signed-rank test across all 100 bootstrap AUC values between the best and second-best methods to test whether the two methods have non-equal performance[53,54]. We use the same 100 test set bootstrap samples for both methods so we can do a paired comparison. The Wilcoxon test is a non-parametric alternative to a paired t-test.

Statistical comparisons of classifiers are a complex topic and frequently subject to mistakes, for example Wilcoxon is appropriate for classifier comparisons whereas a t-test is not[54]. Likewise, we intentionally do not provide confidence intervals here to avoid misinterpretation. First, for non-normal data they tend to provide incorrect coverage[55]. Second, and more crucially, the performance across the bootstrap samples for different methods are non-independent as the variance in performance is driven by the sampling. Thus, overlapping confidence intervals do not imply non-significant differences. If method A performs better than method B for all or a great many of the individual bootstrap samples, then we should conclude that A is better than B, even if A's advantage over B is small relative to the variance between bootstrap samples. This is what the Wilcoxon test captures.

We perform a statistical comparison for each predicted label individually and report statistically significant wins and ties at that level, rather than aggregating them across a whole dataset as such aggregation would lose information and would bring additional statistical complexity. The goal of reporting these wins and ties is to give the reader a summary statistic of how the three models compared. Supplementary S9 has tables with detailed breakdowns of the comparisons between the three main models performed in the main manuscript.

### Reporting summary

Further information on research design is available in the Nature Portfolio Reporting Summary linked to this article.

## Data availability

All datasets used in this manuscript are publicly available. We further make our code, model, and additional information to aid reproducibility available on our GitHub. AIROGS dataset: https://zenodo.org/records/5793241, DDR dataset: https://github.com/nkicsl/DDR-dataset, ODIR-2019 dataset (registration required): https://odir2019.grand-challenge.org, BRSET dataset (registration required): https://physionet.org/content/brazilian-ophthalmological/1.0.0/, IDRiD dataset: https://ieee-dataport.org/open-access/indian-diabetic-retinopathy-image-dataset-idrid, ROP dataset: https://figshare.com/articles/figure/_b_A_Fundus_Image_Dataset_for_Intelligent_b_b_Retinopathy_of_Prematurity_b_b_System_b_/25514449, For the Messidor2, Retina and JSIEC1000 datasets, we used the pre-processed versions provided by the authors of the original RETFound-MEH which can be found here: https://github.com/rmaphoh/RETFound_MAE/blob/main/BENCHMARK.md The RETFound-MEH checkpoints we compare with can also be found at that link. Source data are provided with this paper.

## Code availability

We share the trained RETFound-Green model as well as training and evaluation code on our GitHub: https://github.com/justinengelmann/RETFound_Green

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

## Acknowledgements

We thank the authors of the original RETFound-MEH, Dr Yukun Zhou, Prof. Pearse Keane and colleagues, as well as the authors of DERET-Found, Prof. Yan Bo and colleagues, for their contribution to the field and particularly for making their models openly available which enables the comparisons in this work. We further thank the researchers that made the datasets used in this work available and the individuals who contributed their data to biomedical research. It cannot be emphasised enough that open datasets are an invaluable contribution to progress in our field. J.E. was supported by the United Kingdom Research and Innovation (grant EP/S02431X/1), UKRI Centre for Doctoral Training in Biomedical AI at the University of Edinburgh, School of Informatics. The original work, including the methods development and model training, was done during J.E.'s time at Edinburgh. For part of the manuscript revisions during peer review, J.E. was supported by the Medical Research Council grant MR/Y011651/1. M.O.B. was supported by: Fondation Leducq Transatlantic Network of Excellence (17 CVD 03); EPSRC grant no. EP/X025705/1; British Heart Foundation and The Alan Turing Institute Cardiovascular Data Science Award (C-10180357); Diabetes UK (20/0006221); Fight for Sight (5137/5138); the SCONe projects funded by Chief Scientist Office, Edinburgh & Lothians Health Foundation, Sight Scotland, the Royal College of Surgeons of Edinburgh, the RS Macdonald Charitable Trust, and Fight For Sight.

## Author contributions

J.E. conceptualised the study, developed and implemented the methodology, trained the model, conducted the downstream analyses. J.E. and M.B. analysed the results, wrote, reviewed, edited and revised the manuscript. M.B. provided supervision and guidance.

## Competing interests

The authors declare no competing interests.
