## [Transparent Peer Review file · Nature Communications]

Training a high-performance retinal foundation model with half-the-data and 400 times less compute

Corresponding Author: Professor Miguel Bernabeu

Version 0:

Reviewer comments:

Reviewer #1

(Remarks to the Author)

This paper tackles an interesting and important issue: improving the energy efficiency of AI training for foundation models, specifically for retinal images to reduce environmental impact. However, the results presented are not entirely convincing, primarily because the experiments conducted don't sufficiently support the claims made.

The authors evaluated their proposed model on only two datasets: BRSET and IDRiD. This narrow focus doesn't provide comprehensive evidence of the model's efficacy. Additionally, the paper lacks a comparison with traditional supervised learning approaches, which typically involve splitting the downstream dataset into training and test sets to evaluate performance (without pre-training).

Foundation models are valued for their ability to perform well on diverse datasets and downstream tasks, even with limited data available for fine-tuning. However, the authors used 80% of the BRSET dataset (16,266 fundus images) for training. As a result, it's not surprising that their model does not perform systematically worse on the remaining data from the same dataset. This approach undermines the primary advantage of foundation models: their adaptability to a variety of tasks.

The proposed token reconstruction technique, as opposed to image patch reconstruction, is intriguing. However, the paper does not clearly explain how this method reduces training time, which is critical to the paper's energy-efficiency argument.

To better demonstrate the effectiveness of their approach compared to other foundation models, the authors should validate their method on diverse datasets and a wider range of downstream tasks. Internal and external testing would also be beneficial to prove the model's robustness across different data sources and scenarios.

(Remarks on code availability)

Reviewer #2

(Remarks to the Author)

This manuscript describes a more efficient approach to training foundation models for retinal color fundus photographs. Although the work is well motivated ("green", better for low compute resource settings) and has the potential to be a valuable tool for the community, from a technical perspective there are a few questions about the comparisons to baselines and potential ablation studies that are missing

Minor:

-- "MAE involves pixel-level re-construction of images which is effective for the high-level diversity in natural images, not well suited for capturing small structures." Please cite a source for this assertion

--"propose a novel Token Reconstruction self-supervised learning strategy that focuses on higher-level, abstract features

and is designed for making domain-specific foundation models instead of training models from scratch for general computer vision." Can you explain this in the context of the previous statements that the goal is to focus on small structures vs. abstract concepts?

-- "is designed for making domain-specific foundation models instead of training models from scratch for general computer vision." - please clarify these terms in the context of your work. "domain specific foundation models", "Training from scratch" Why wouldn't a model trained from scratch on retinal images be considered domain specific? Where does SSL fine-tuning using DINO v2 weights fall in your definition of domain specific?

-- The concerns about Python 3.7.5 for RETfound-MEH are a bit misplaced perhaps? It is not surprising that technology evolves and exact versions of python become deprecated. Most people have not found it challenging to get RETfound-MEH to work and/or port it as needed.

-- Can you confirm that all other factors were kept consistent across the two experiments (data normalization, image sizes, cropping/padding/resizing techniques) ?

-- "uses the original vision transformer architecture [27] but with four extra "register tokens" - Would suggest adding ablation studies showing performance without and with using these 4 extra "register tokens" as well as the other token strategies for group tokenization.

-- "RETFound-Green to process the images at higher resolution of 392x392 instead of 224x224 used by the other two models while still being more efficient at the same time." -for the comparisons to be fair, would suggest adding performance of using 224x224 for RETFound-Green

--"only use the DinoV2 weights and then train those on retinal images using our own Token Reconstruction strategy, described in the next section, and our own codebase." - a comparison of what happens when you start with MAE weights would be helpful for a more balanced comparison

-- please comment on the impact of the adding more tokens in terms of reconstruction

-- "We detect the fundus area and then crop horizontal black space, or pad the top and bottom, to ensure the images are properly cropped and square before resizing. We implement this because we think it is a principled way to resizing retinal images, but we do not think that this has any substantial impact at all on performance." Please provide more details here and evidence to back these statements with additional ablation studies.

-- "For downstream evaluations, we simply resize the images to the desired size and normalise by each model's specific normalisation constants. We intentionally do not use our more complex preprocessing on the downstream tasks to make the comparisons fair by ensuring that better preprocessing plays no role." -shouldn't this principle be applied to the SSL as well? Keep everything the same except for the use of your novel architecture/approach?

(Remarks on code availability)

Reviewer #3

(Remarks to the Author)

This paper introduces an efficiency-driven eye foundation vision model. The primary methodology involves a reconstruction pretraining approach rather than a mask prediction pretraining approach. In simple terms, when provided with an image, the model initially employs a frozen image encoder to generate a feature representation. Then, it introduces noise and corrupts patches, using a trainable image encoder to obtain a feature representation (the reconstructed version), which is further compared with the original feature representation. The study claims that this proposed approach demonstrates greater efficiency, requires less training data, and has superior performance compared to two baseline models.

Main comments:

The three models (the proposed model and two baselines) are pretrained on entirely different datasets, thus not enabling a direct comparison. At the very least, the study should use the same pretraining dataset and adopt the mask prediction pretraining approach as the baseline to assess performance trade-offs.

Furthermore, the study fails to compare with any efficiency-based approaches. The mask prediction pretraining approach, being the default method, does not prioritize efficiency. Numerous efficiency-driven approaches have been proposed, such as pruning-based and distillation-based techniques, which could significantly enhance efficiency compared to the default

approach.

Another crucial aspect is the limited evaluation of zero/few-shot capabilities. A foundational aspect of a model is its ability to learn a generic representation adaptable to downstream tasks, even for out-of-distribution scenarios. However, the benchmarks predominantly cover diseases included in the pretraining set of the proposed approach. The downstream accountability is not clear.

(Remarks on code availability)

It only has example codes and does not have pretraining, fine-tuning, and inference codes.

Reviewer #4

(Remarks to the Author)

(Remarks on code availability)

Reviewer #5

(Remarks to the Author)

(Remarks on code availability)

Version 1:

Reviewer comments:

Reviewer #1

(Remarks to the Author)

Thank you to the authors for including a new unseen dataset and additional experiments. However, these additions are not entirely convincing in demonstrating the generalizability of the foundation model without external testing on more unseen datasets. While it is understood that the three unseen datasets were not used for pre-training, external testing remains crucial.

External testing involves fine-tuning the model on one dataset for a specific task and evaluating its performance on the same task using a new dataset from a different cohort, medical center, or ethnic group. Such testing would provide valuable evidence of the model's robustness across diverse data sources and scenarios. Without validation of the foundation model in this setting, it may be difficult to substantiate the claims on the benefits of foundation model.

(Remarks on code availability)

Reviewer #2

(Remarks to the Author)

thank you for addressing the issues raised

(Remarks on code availability)

Reviewer #3

(Remarks to the Author)

Thanks for the substantial efforts on the revision. Many of my comments have been addressed; however, some of my concerns regarding the evaluation remain.

The study mentions that the proposed model achieved the best performance in 68 tasks out of 119 comparisons. What exactly is considered a "task"? Are the main classifications in Figure 6 considered one task, with their sub-analyses considered separate tasks? Since the evaluation was conducted on three datasets, it is important to provide details on the exact tasks, including the settings (e.g., train/test or external), and the number of instances.

Additionally, the evaluation is limited compared to the baseline models such as RETFound: (1) Only two primary eye disease severity-level classifications were evaluated. Other eye diseases were evaluated as binary classifications (yes/no), which is easier and less clinically useful; (2) Only classification tasks were evaluated, without including other tasks such as progression prediction; (3) Although the study compared its model with RETFound, it did not use any of RETFound's testing sets for direct comparison, nor did it directly cite RETFound's performance.

(Remarks on code availability)

Version 2:

Reviewer comments:

Reviewer #3

(Remarks to the Author)

Thanks for the thoughtful revision. This version addresses my main concerns.

(Remarks on code availability)

Reviewer #6

(Remarks to the Author)

The article is much clearer after the revisions, and the authors have developed an efficient retinal foundation model. However, at this stage, the downstream tasks for the ophthalmology-specific model mostly rely on publicly available datasets. There are still challenges in applying this model to the diverse and dynamic clinical setting of ophthalmic diagnoses and follow-ups. It would be valuable for the authors to consider expanding the scope of their work by exploring the potential for deploying this efficient model in real-world clinical environments, including addressing the variability in patient populations and clinical workflows. A discussion on future directions for integrating this model into clinical practice would be a significant contribution.

(Remarks on code availability)

Reviewer #1 (Remarks to the Author):

This paper tackles an interesting and important issue: improving the energy efficiency of AI training for foundation models, specifically for retinal images to reduce environmental impact. However, the results presented are not entirely convincing, primarily because the experiments conducted don't sufficiently support the claims made.

We thank the reviewer for taking the time to review our manuscript and for providing their comments, which helped us substantially improve our manuscript. In particular, we have substantially expanded the experimental results in response to the comments by the other reviewers and your own.

In the original manuscript, we compared the three foundation models on their adaptability for downstream classification across two unseen datasets (BRSET and IDRiD) on 29 tasks.

In the revised manuscript, we now compare the three foundation models for:

- Downstream classification across *three* unseen datasets on 43 tasks, adding a Chinese dataset of images of young infants relating to Retinopathy of Prematurity (ROP) and updating our diabetic retinopathy tasks, e.g. considering both the whole of BRSET and BRSET subjects with diabetes only (the latter mirroring a typical diabetic retinopathy screening programme).
- Unsupervised lower-dimensional projections of the feature vectors using both Principal Component Analysis and UMAP for the ROP and IDRiD datasets, qualitatively and quantitatively examining whether the projections (which are obtained entirely without disease labels) separate images by disease status.
- Transportability of feature vectors, using diabetic retinopathy classifiers fit to IDRiD and evaluated on BRSET, and vice versa.

Thus, we not only substantially expanded our original comparisons but added two additional types of comparisons, too. Furthermore, we now also provide results for a RETFound-Green model trained at a lower resolution of 224x224 pixels, denoted as RETFound-Green@224. Finally, we provide additional experiments where relevant in the supplementary.

The new results paint a picture consistent with our original results. RETFound-Green performs the best overall (68 statistically significant wins across 119 comparisons), suggesting that it does not perform systematically worse and possibly generally better than the other two models.

We have made a few minor changes as well. For example, BRSET provides diabetic retinopathy grades using two systems: the Scottish Diabetic Retinopathy Grading Scheme and the International Clinic Diabetic Retinopathy Scale. Both are highly similar, and in the original manuscript we used the Scottish one for no strong reason other than us being based in Scotland. However, IDRiD uses the international scale. Thus, for consistency and to enable the

transferability experiments, we now use the international scale throughout. This minorly changed some of the results, but without introducing any meaningful qualitative changes.

Please see our point-by-point replies to your own comments below, and please also consider our responses to the other reviewers where they might be relevant.

The authors evaluated their proposed model on only two datasets: BRSET and IDRiD. This narrow focus doesn't provide comprehensive evidence of the model's efficacy.

We thank you for this comment, which we completely agree with. While the results in our original manuscript were very encouraging, they could have been more comprehensive.

Based on your comment, we added an additional dataset, namely the ROP dataset from China. This dataset is particularly interesting, given that – unlike all of the other downstream datasets, as well as all of the pre-training datasets – it contains images of young infants. Additionally, as mentioned above, we have substantially expanded our experiments, looking at unsupervised lower-dimensional projections of the feature vectors and transportability across datasets.

The additional experiments paint a picture consistent with our original results and we think there is now comprehensive evidence that our model is not systematically worse than the other two models – despite using half-the-data and 400 times less compute – and perhaps even has a slight edge in terms of performance.

Additionally, the paper lacks a comparison with traditional supervised learning approaches, which typically involve splitting the downstream dataset into training and test sets to evaluate performance(without pre-training).

The three downstream datasets (BRSET, IDRiD, ROP) were not used for pre-training of any of the three foundation models. As you suggest, the downstream datasets were split into training and test sets. IDRiD provides an official train-test split which we use. For BRSET and ROP, we split them into training and test sets at the patient level, i.e. if there are multiple images of the same patient, they all appear in the same set. This avoids data leakage by ensuring that there is no overlap in patients between the training and test set.

In the updated manuscript, we further augment the downstream classification results by additionally examining unsupervised two-dimensional projections of the feature vectors.

To make this clearer, we have split the Methods section “Datasets for pre-training and downstream evaluations” into two separate sections “Datasets for pre-training” and “Datasets for downstream evaluations”.

For the “Datasets for pre-training” section, we now specify explicitly: *“None of these pre-training datasets were used in any of the downstream evaluations.”*

Likewise, for the “Datasets for pre-training” section, we now clarify after describing the train-test splitting: *“For the avoidance of doubt, no part of these downstream datasets were used in the pre-training of any of the three foundation models we compare.”*

We think that these changes have made the manuscript more precise and less ambiguous.

Foundation models are valued for their ability to perform well on diverse datasets and downstream tasks, even with limited data available for fine-tuning. However, the authors used 80% of the BRSET dataset (16,266 fundus images) for training. As a result, it's not surprising that their model does not perform systematically worse on the remaining data from the same dataset. This approach undermines the primary advantage of foundation models: their adaptability to a variety of tasks.

To avoid any potential misunderstanding, none of the downstream datasets were used for pre-training of our model. All downstream datasets are completely unseen to all three models. That includes BRSET. If the pre-training data of our model included any of the downstream datasets, that would be a serious issue indeed. However, that is not the case.

Furthermore, the exact same procedure and data splits are used to adapt all three foundation models, so that the comparison is like-for-like and fair.

To clarify this point, in the updated manuscript, we added a new Supplementary S1 that compares the pre-training datasets of the three models. Note that the pre-training data for RETFound-Green is a strict subset of the pre-training data for DERETFound.

	RETFound-MEH	DERETFound	RETFound-Green
Datasets	Moorfields Diabetic imAge dataSet (MEH-MIDAS)*, Kaggle EyePACS diabetic retinopathy	AIROGS, DDR, ODIR-2019, Kaggle EyePACS diabetic retinopathy	AIROGS (subsample)*, DDR, ODIR-2019
Notes	* MEH-MIDAS is not openly available.	/	*53,327 randomly selected images, 46.8% of the dataset.
Total images	904,170	150,786	75,000

Supp. Table 1: Overview of the datasets used for pre-training.

Inspired by your comment pointing to the size of BRSET, we also briefly investigate performance on BRSET using only a fraction of the training set for downstream adaptation. This is only a small sensitivity analysis and not completely exhaustive: We sample 10 and 20 positive cases per label once and additionally consider having 100 randomly selected negative cases and having all available negative cases for adaptation.

The results are reported in Supplementary S5 and generally consistent with our main results. Indeed, at 10 positive cases, RETFound-Green achieves slightly more wins than in the main experiments with all of the BRSET training set. As only the linear probe is fit to these images, this should in theory favour models with higher dimensional feature vectors, as this gives more degrees of freedom that can make use of this amount of downstream training data. Thus, the original setup favoured the other two models over our own.

We would also like to point out that a benefit of foundation models and linear probing is faster prototyping and a lower bar for knowledge of deep learning compared to traditional finetuning. Fitting a logistic (or linear) regression from pre-computed feature vectors is both computationally very efficient and easier to implement. Thus, in our opinion, even in settings with 16,000 training images, foundation models are relevant and the original experiments informative.

The proposed token reconstruction technique, as opposed to image patch reconstruction, is intriguing. However, the paper does not clearly explain how this method reduces training time, which is critical to the paper's energy-efficiency argument.

We thank the reviewer for pointing out that this needs to be explained better. Briefly, we use a different self-supervised learning objective that we think is much more suitable and this is what allows our model to achieve good performance in a much more efficient way.

First, the Masked AutoEncoder (MAE) objective trains an additional reconstruction model which is discarded after the pre-training, and the output of the MAE pipeline is a full image, whereas our objective only has a feature vector as output. The additional model and larger output both increase computational complexity and thus avoiding them makes our method more efficient.

Second, since our objective is focused on reconstruction in an abstract feature space instead of pixel space, the model is encouraged to learn a salient compressed representation unlike MAE which is encouraged to retain sufficient information for pixel-level reconstruction. This difference allows us to use a smaller model while retaining very good performance.

The combination of not needing an auxiliary decoder model and a smaller model means that each training iteration is much less compute- and time-intensive. Finally, our objective allows our model to get good performance with much fewer training iterations. The combination of much fewer iterations and much cheaper iterations leads to order-of-magnitude improvements. Of course, this is only useful if our token reconstruction objective is indeed effective and yields a high-performance number. This is what we are showing in the manuscript through our experiments.

We have now clarified this in the “Computational resources” section: *“Our approach is much more compute efficient during training as we train a smaller model without an additional decoder for fewer image-iterations, yet our Token Reconstruction objective allows us to achieve high performance nonetheless.”*

To better demonstrate the effectiveness of their approach compared to other foundation models, the authors should validate their method on diverse datasets and a wider range of downstream tasks.

We completely agree. Please see our earlier response to your first comment regarding newly added additional datasets and evaluations. Our updated experiments are substantially expanded and provide further evidence for the effectiveness of our approach.

Internal and external testing would also be beneficial to prove the model's robustness across different data sources and scenarios.

As mentioned above, none of the downstream datasets were used for training. So in a sense our experiments already evaluate how the models perform on different data sources.

However, we completely agree that robustness across different data sources and scenarios is very important, especially in medical AI. Thus, we address your comments by additional experiments in two distinct ways:

First, as mentioned above, we have added the ROP dataset. This covers a condition not present in any of the other datasets (ROP), and a demographic also not present in any of the other datasets (young infants). Thus, this is quite a different scenario.

Second, we now explicitly evaluate how well classifiers trained on the feature vectors of the three foundation models transfer from one downstream dataset to another. As an exemplar, we use diabetic retinopathy screening and grading, as this is a very common and fairly well standardised task. We look at BRSET and IDRiD and fit the classifier on one dataset and evaluate on the other. The results are reported in the new section “Transportability of feature vectors across datasets” and Fig. 5. Additionally, in Supplementary S3, we conduct a similar experiment using the two-dimensional projections obtained with PCA and UMAP.

In these new experiments, we find that the three models, including our own, are indeed robust to different scenarios and that classifiers trained on foundation model feature vectors do transfer from one dataset to another.

We think that these newly added experiments are an important and very informative addition to our work, so we really appreciate the suggestion.

Reviewer #2 (Remarks to the Author):

This manuscript describes a more efficient approach to training foundation models for retinal color fundus photographs. Although the work is well motivated ("green", better for low compute resource settings) and has the potential to be a valuable tool for the community, from a technical perspective there are a few questions about the comparisons to baselines and potential ablation studies that are missing

We thank the reviewer for taking the time to review our manuscript and for providing their comments, which helped us substantially improve our manuscript. In particular, we have substantially expanded the experimental results in response to the comments by the other reviewers and your own.

In the original manuscript, we compared the three foundation models on their adaptability for downstream classification across two unseen datasets (BRSET and IDRiD) on 29 tasks.

In the revised manuscript, we now compare the three foundation models for:

- Downstream classification across *three* unseen datasets on 43 tasks, adding a Chinese dataset of images of young infants relating to Retinopathy of Prematurity (ROP) and updating our diabetic retinopathy tasks, e.g. considering both the whole of BRSET and BRSET subjects with diabetes only (the latter mirroring a typical diabetic retinopathy screening programme).
- Unsupervised lower-dimensional projections of the feature vectors using both Principal Component Analysis and UMAP for the ROP and IDRiD datasets, qualitatively and quantitatively examining whether the projections (which are obtained entirely without disease labels) separate images by disease status.
- Transportability of feature vectors, using diabetic retinopathy classifiers fit to IDRiD and evaluated on BRSET, and vice versa.

Thus, we not only substantially expanded our original comparisons but added two additional types of comparisons, too. Furthermore, we now also provide results for a RETFound-Green model trained at a lower resolution of 224x224 pixels, denoted as RETFound-Green@224. Finally, we provide additional experiments where relevant in the supplementary.

The new results paint a picture consistent with our original results. RETFound-Green performs the best overall (68 statistically significant wins across 119 comparisons), suggesting that it does not perform systematically worse and possibly generally better than the other two models.

We have made a few minor changes as well. For example, BRSET provides diabetic retinopathy grades using two systems: the Scottish Diabetic Retinopathy Grading Scheme and the International Clinic Diabetic Retinopathy Scale. Both are highly similar, and in the original manuscript we used the Scottish one for no strong reason other than us being based in Scotland. However, IDRiD uses the international scale. Thus, for consistency and to enable the

transferability experiments, we now use the international scale throughout. This minorly changed some of the results, but without introducing any meaningful qualitative changes.

Please see our point-by-point replies to your own comments below, and please also consider our responses to the other reviewers where they might be relevant.

Minor:

-- "MAE involves pixel-level re-construction of images which is effective for the high-level diversity in natural images, not well suited for capturing small structures." Please cite a source for this assertion

This statement is merely meant to convey an intuition for why our approach might be advantageous, so we have revised the passage in question to convey uncertainty.

"MAE involves pixel-level re-construction of images which is effective for the high-level diversity in natural images, but might not be well-suited for capturing small structures."

While there is no space in the introduction to go into more detail, we added the following detailed discussion in the methods section to elaborate on this:

"Though we do not investigate it in detail in this work, we think that the fact that our method reconstructs patches in a perceptual, abstract space as opposed to pixel space is a key advantage. MAE models trained to minimise mean squared error tend to produce blurry reconstructions, as can be seen in both the MAE and RETFound-MEH/DERETFound papers. This is because the optimal solution for minimising mean squared error is to predict the conditional expectation of the masked patches y given the non-masked patches x , $E[y|x]$. Intuitively, this expectation is a weighted average of all possible reconstructions of the partially masked image, each possible reconstruction weighted by how likely it is. Averaging over all possible reconstructions produces a blurry image.¹

With our Token Reconstruction approach in the other hand, the prediction target is a feature vector providing an abstract description of the input patch. Return to our example above [in the paragraph above this quote in the relevant methods section in the manuscript] of a masked patch being highly likely to contain small diabetic lesions. Their exact shape and location will be hard to predict, and averaging over all possible reconstructed patches with lesions as MAE encourages will yield a uniform patch with the background retina hue. Thus, encoding the presence of lesions would not substantially improve the loss over just encoding the background hue. Whereas for our Token Reconstruction objective, the optimal prediction will

¹ The official MAE GitHub repository provides a model trained without their pixel normalisation, which yields better-looking visualisations but worse representations. They even provide an additional model trained with an adversarial loss to produce more realistic and sharper images (<https://github.com/facebookresearch/mae?tab=readme-ov-file#visualization-demo>) for visualisation purposes, with the adversarial loss forcing the reconstruction towards a particular possible reconstruction. While not discussed in the paper, presumably the adversarial loss also comes at the cost of yielding worse representations for downstream adaptations of the model.

be a weighted average of all abstract representations of the masked patch. We conjecture that the average over all abstract representations of many possible patches with small lesions will be a vector that still captures the presence of such patterns.

For natural image datasets like ImageNet, there is a lot of high-level variation in the images, which in the machine learning community are referred to as “low-frequency patterns”. Consider two images of cars. The cars might not only be different makes and colours, but the images might be taken in different lighting conditions, from different angles, with different amounts and kinds of backgrounds. Now consider that ImageNet contains very diverse classes of objects, from vehicles to specific dog breeds to specific foods. Even a blurry reconstruction of such diverse images requires that the MAE encoder captures key details relevant to the class of the input image. In retinal imaging, on the other hand, all images are comparatively highly similar and very small structures such as a few small lesions can be key to differentiating a healthy retina from an unhealthy one. Thus, we think that doing the reconstructions in an abstract space is a key advantage in retinal imaging. Future work should investigate this in more detail, for example by adapting the different retinal foundation models for segmentation.”

In informal discussions we have had with colleagues, the notion that auto-encoders trained with mean squared error lead to blurry reconstructions is quite well accepted, but we do not have a concrete source to point to, so we have reduced the certainty of our original statement and added the elaboration.

--"propose a novel Token Reconstruction self-supervised learning strategy that focuses on higher-level, abstract features and is designed for making domain-specific foundation models instead of training models from scratch for general computer vision." Can you explain this in the context of the previous statements that the goal is to focus on small structures vs. abstract concepts?

Please see the prior response. For datasets with a lot of high-level diversity in the images (=low-frequency patterns), small structures are less important than in retinal imaging. Our approach aims to reconstruct abstract, perceptual representations of the patches, so the presence of small “high-frequency” patterns will still be picked up, whereas in the case of MAE such small structures will not be well-reconstructed by the decoder and thus it is not beneficial for the encoder to learn to capture their presence.

-- "is designed for making domain-specific foundation models instead of training models from scratch for general computer vision." - please clarify these terms in the context of your work. "domain specific foundation models", "Training from scratch" Why wouldn't a model trained from scratch on retinal images be considered domain specific? Where does SSL fine-tuning using DINO v2 weights fall in your definition of domain specific?

First, we would like to note that all three foundation models make use of pre-trained SSL weights from natural image datasets that are then fine-tuned (with SSL) on retinal images. In the case of RETFound-MEH and DERETFound, they use the official MAE weights and the MAE

SSL method to fine-tune them. We use DinoV2 weights and fine-tune them with our own Token Reconstruction approach. We now make this more explicit:

*“RETFound-MEH and DERETFound start with pre-trained weights from the Masked Autoencoder (MAE) paper [22] and then train these weights on retinal images using the MAE strategy and codebase. We start with weights from DinoV2 [43], a self-supervised learning method like MAE. However, unlike the other two foundation models, we only use the DinoV2 weights and then train those on retinal images using our own Token Reconstruction strategy, described in the next section, and our own codebase. **The DinoV2 self-supervised learning approach or codebase are not used.**”*

Indeed, it would be perfectly possible to train a domain-specific model for retinal images from scratch, i.e. without re-using prior weights. Conversely, it would equally be perfectly possible to training a general computer vision model for natural images “not-from scratch” through SSL fine-tuning of weights that have been trained on other data, e.g. weights that had initially been trained on retinal images.

What we want to emphasize here is that MAE was developed in a specific context, namely training a general computer vision model from scratch. This context differs substantially from the context at hand, namely training a domain-specific foundation model for retinal images. Natural images and retinal images are quite different structures. And for developing a retinal foundation model, we do not need to train from scratch (indeed, both RETFound-MEH and DERETFound make use of pre-trained weights already). Thus, it is quite plausible that MAE is not optimal for the task at hand and another approach such as ours could deliver substantial benefits.

-- The concerns about Python 3.7.5 for RETfound-MEH are a bit misplaced perhaps? It is not surprising that technology evolves and exact versions of python become deprecated. Most people have not found it challenging to get RETfound-MEH to work and/or port it as needed.

Thanks for your feedback. Based on your input, we have de-emphasized this point by making the relevant passage more concise and moving half of it to a footnote.

We agree that porting RETFound-MEH to newer versions of Python/PyTorch/timm would be feasible, though it requires a higher level of technical knowledge. We also agree that technology evolves and software “decays” without active effort to maintain it. However, we would like to add to consideration the fact that the RETFound-MEH paper was published only a year ago and Python 3.7.5 was deprecated earlier this year, so this state of affairs did not only materialise a long time after the work was initially published.

In most cases, installing an old version of Python is feasible, though we think that especially in medical imaging, working in safe havens is not uncommon where this might not be possible. Similarly, needing to port the model weights raises the bar for technical knowledge required. In the anecdote we’re describing, a colleague who is a post-doctoral researcher with many years

of experience in machine learning-based medical image analysis who regularly uses deep learning in their research did not know how to do so. We think that this is not an exception, but instead we assume that there would be many researchers like our colleague.

Thus, we agree that this point does not need to be overstated and addressed your comment by shortening the passage and additionally moving some of it to a footnote.

-- Can you confirm that all other factors were kept consistent across the two experiments (data normalization, image sizes, cropping/padding/resizing techniques) ?

All experiments were designed to be as comparable and fair as possible.

For image normalisation, we used the “correct” parameters (channel-wise “means” and “standard deviations” for centering and scaling of the data) for each model, i.e. those that the given model was trained with and that are thus recommended for its downstream use.

For image sizes, we used the “correct” image size for each model, i.e. the one the model was trained with and that is thus recommended for its downstream use. Please note that based on your comments, we also trained a RETFound-Green model at 224x224 resolution. Please see our reply to your relevant comment further down below.

For cropping/padding/resizing, we used simple resizing, which is what is used by the other works we compare with. When we pre-processed the data for our own pre-training, we first cropped the black areas, padded to square, and then resized. However, for a fair comparison – or rather on that intentionally errs on the side of being unfavourable to our own model -, we used the same simple resizing approach for all models. In other words, we used the “correct” preprocessing for the baseline models and a (slightly) out of distribution one for our own model.

Based on your feedback, we have updated the passage in question to the following:

*“We implement this **purely** because we think it is a **more principled way to resize** retinal images in a way that preserves the aspect ratio. In principle, this resizing approach could offer an advantage in downstream performance. In our opinion, this is unlikely, especially given that we conducted our downstream evaluations without using this pre-processing approach.*

For downstream evaluations, we simply resize the images to the desired size and normalise by each model’s specific normalisation constants. Thus, we use the same preprocessing for all three models during the downstream evaluations. This matches how the images were resized for RETFound-MEH and DERETFound to ensure a fair comparison. We intentionally do not use our more complex preprocessing on the downstream tasks to make the comparisons fair by ensuring that better preprocessing plays no role.”

-- "uses the original vision transformer architecture [27] but with four extra “register tokens” - Would suggest adding ablation studies showing performance without and with

using these 4 extra "register tokens" as well as the other token strategies for group tokenization.

We agree that this is an interesting question. In our opinion, this does not majorly change the interpretation of our results, as we aim to present a more efficient yet effective approach, and choosing this slight architectural modification does not add any additional complexity to the implementation of our approach. Thus, we chose this version as it is a priori slightly preferable (based on the findings reported by Darcet et al. 2023 *Vision Transformers Need Registers*) with no additional effort in implementation and very negligible computation cost. However, unlike the experiment you suggest in your next comment, we do not think that this ablation would meaningfully change the interpretation of our results.

In our discussion, we now mention that additional experiments on this would be interesting:

“Fourth, future work should develop our Token Reconstruction method further and investigate the impact of design choices such as model architecture, original pre-training method, or parameter count.”

-- "RETFound-Green to process the images at higher resolution of 392x392 instead of 224x224 used by the other two models while still being more efficient at the same time." - for the comparisons to be fair, would suggest adding performance of using 224x224 for RETFound-Green

This is a very important point raised by the reviewer, that we completely agree with. We would like to note that the input resolution is a design choice and improvements driven by using a higher resolution would still be genuine improvements. For the other two models, increasing the resolution would drastically increase the amount of compute required for pre-training (possibly necessitating a smaller batch size as well), and also make them less efficient for inference. Part of the advantage of our approach is that it is far more efficient while – at the same time – using a higher resolution.

Thus, in a sense our comparisons are already fair: Our model is cheaper to train and faster at inference time, despite the increased resolution. From a practitioner’s perspective, RETFound-Green is a better choice. Training the other two models with higher resolution images would be non-trivial and expensive and they would be even less efficient than our model at inference time.

However, we agree this is a very important and interesting question: Is this increase in resolution the sole reason why RETFound-Green performs so well, despite being trained using much less data and compute?

Thus, we trained a RETFound-Green model at a resolution of 224x224 (“RETFound-Green@224”) and added the suggested experiments. The results can be found in Figure 6 of the updated manuscript. Briefly, RETFound-Green@224 still performs well and even achieves the most statistically significant wins on the classification tasks when compared with

RETFound-MEH and DERETFound. The performance is slightly worse than our proposed RETFound-Green using 392x392 as resolution.

This suggests that our approach is also effective at the lower resolution and it is not solely the increase in resolution that allows our model to perform well. Additionally, there is a benefit to the increased resolution, so our original design choice of using a higher resolution was also useful.

We paste the relevant figure below for convenience, and some additional relevant discussion in the manuscript.

a) ROP (China) – Retinopathy of Prematurity

b) BRSET (Brazil) – Diverse tasks

c) IDRiD (India) – Diabetic Retinopathy

d) BRSET (Brazil) – Diabetic Retinopathy

RETFound-MEH (red), DERETFound (orange), RETFound-Green@224 (green)

Figure 6: Performance across a variety of models and tasks, using a RETFound-Green model trained and evaluated at the same 224x224 resolution as the other two models. For robustness, reported results are the median of 100 bootstrap samples of the test set. Best result for each task in bold, the bar with p-value indicates the result of a Wilcoxon signed-rank test between the best and second best methods across the 100 bootstrap samples, with $p < 0.05$ in bold. “***” indicates $p < 0.0001$.

--"only use the DinoV2 weights and then train those on retinal images using our own Token Reconstruction strategy, described in the next section, and our own codebase." - a comparison of what happens when you start with MAE weights would be helpful for a more balanced comparison

We agree that there are many interesting questions to investigate. However, in our opinion this particular experiment – while interesting – goes beyond the scope of our current manuscript. We explicitly designed all the experiments with fairness in mind, erring on the side of biasing them against our model. However, our model also used half-the-data and 400 times less compute, and – as our newly added experiments based on your suggestion show – our approach is effective even when lowering our resolution. Thus, we believe that our comparison as presented here is already balanced and allows for a meaningful interpretation of our results.

As mentioned above, we think this would be interesting to explore in the future and now mention it in the discussion:

“Fourth, future work should develop our Token Reconstruction method further and investigate the impact of design choices such as model architecture, original pre-training method, or parameter count.”

-- please comment on the impact of the adding more tokens in terms of reconstruction

There are two principal ways to add more tokens to the model: First, increasing the number of tokens relating to spatial positions, i.e. increasing the number of patches. This could be through increasing the resolution, decreasing the patch size, or a combination of the two. This might allow the model to capture more fine-grained information, but the computational complexity of a ViT model is quadratic in the number of patches due to the attention layers, and thus more patches quickly increases the computational burden. There is another effect to consider: As the patch size changes the pattern of corrupted patches also changes, thus implicitly changing the task – e.g. corrupting 25% of patches in a 2x2 grid is characteristically different from corrupting 25% of patches in a 100x100 grid. Future work could investigate this and we note that we could sample neighbouring blocks of patches instead of individual patches to modulate this effect independently of the patch size.

Second, in the context the previous question, the register tokens are additional tokens that need to be reconstructed. These tokens are meant to serve as registers that provide additional capacity for internal computation. Reconstructing those tokens does not meaningfully increase computational burden and likely encourages the model to capture the global context of the image better, due to these tokens not relating to a specific spatial position.

We have now expanded the relevant discussion in the manuscript:

“Like RETFound-MEH and DERETFound, RETFound-Green uses the original vision transformer architecture [27] but with four extra “register tokens” [40] that do not relate to a specific patch. This is a minor and straightforward modification of the architecture that has been observed to

yield smoother attention maps by providing a register for the model to capture additional global context. They add negligible computational cost and reconstructing them might help our model to better capture global context.”

-- "We detect the fundus area and then crop horizontal black space, or pad the top and bottom, to ensure the images are properly cropped and square before resizing. We implement this because we think it is a principled way to resizing retinal images, but we do not think that this has any substantial impact at all on performance." Please provide more details here and evidence to back these statements with additional ablation studies.

The highlighted passage contains two claims: First, that this way of resizing the images – in a way that preserves aspect ratio – is principled. We do not think that this is particularly controversial statement as it avoids distorting the image, nor is it one that we could provide any ablation for as it is a purely conceptual claim, namely that avoiding distortions is principled. Second, that we think that this is unlikely that this has any substantial impact at all on performance. This is a statement that could be tested empirically, but we do not think it is important enough to warrant training another model. Please especially consider the fact that we intend to refrain from using this pre-processing during the downstream evaluations to avoid biasing the results in favour of our model.

We also intentionally signpost these two claims as our opinions (“we think”) in recognition of the fact that we do not provide supporting evidence. In our opinion, preserving the aspect ratio of images is indeed generally desirable from a first principles point of view. We think that it is also unlikely to make a substantial difference in general, but especially given that we only applying this for pre-training. In our opinion, this would if anything disadvantage our method. On both of those points, opinions can of course differ, but we do not consider our position to be particularly unreasonable.

To address this and your previous comment, we have changed the passage in question as follows:

*“We implement this **purely** because we think it is a **more principled way to **resize**** retinal images **in a way that preserves the aspect ratio**. **In principle, this resizing approach could offer an advantage in downstream performance**. **In our opinion, this is unlikely, especially given that we conducted our downstream evaluations without using this pre-processing approach.**”*

*For downstream evaluations, we simply resize the images to the desired size and normalise by each model’s specific normalisation constants. **Thus, we use the same preprocessing for all three models during the downstream evaluations. This matches how the images were resized for RETFound-MEH and DERETFound to ensure a fair comparison.** We intentionally do not use our more complex preprocessing on the downstream tasks to make the comparisons fair by ensuring that better preprocessing plays no role.”*

-- "For downstream evaluations, we simply resize the images to the desired size and normalise by each model's specific normalisation constants. We intentionally do not use our more complex preprocessing on the downstream tasks to make the comparisons fair by ensuring that better preprocessing plays no role." -shouldn't this principle be applied to the SSL as well? Keep everything the same except for the use of your novel architecture/approach?

As discussed in response to your previous comment, we think that the difference in pre-processing during the SSL part is unlikely to substantially change performance. Furthermore, we intentionally design the downstream experiments to make the comparison maximally fair and possibly biased against the method we propose, including using the preprocessing that the other two models used for downstream evaluations, rather than using our complex preprocessing that only our model was trained with.

Our primary goal is to present a new high-performance retinal foundation model and an approach that allows training such a model far more efficiently. Thus, we opted for the approach that we consider more principled to train our model, even if we do not think that this will meaningfully change performance.

More broadly, based on one of your earlier comments, the updated manuscript now contains results for a newly trained RETFound-Green@224 that uses the same resolution as the other two models. The resolution the model operates on is a major detail and thus we completely agree with you that this warranted additional experiments.

We would also like to note that we use a strict subset of the training data used by DERETFound, which provides a direct comparison in terms of pre-training data. This is now clarified in the manuscript, including with the newly added Supplementary S1:

	RETFound-MEH	DERETFound	RETFound-Green
Datasets	Moorfields Diabetic imAge dataSet (MEH-MIDAS)*, Kaggle EyePACS diabetic retinopathy	AIROGS, DDR, ODIR-2019, Kaggle EyePACS diabetic retinopathy	AIROGS (subsample)*, DDR, ODIR-2019
Notes	* MEH-MIDAS is not openly available.	/	*53,327 randomly selected images, 46.8% of the dataset.
Total images	904,170	150,786	75,000

Supp. Table 1: Overview of the datasets used for pre-training.

"Supp. Table 1 gives an overview of the datasets used for pre-training for the three models. RETFound-MEH and DERETFound only overlap regarding the Kaggle EyePACS diabetic

retinopathy dataset, which can be found at: <https://www.kaggle.com/c/diabetic-retinopathy-detection/data>. The MEH-MIDAS dataset is not openly available.

DERETFound and RETFound-Green have substantial overlap, with RETFound-Green's pretraining data being a subset of DERETFound's. In particular, RETFound-Green used one fewer datasets, and only half of the AIROGS dataset.

Overall, RETFound-Green used slightly less than half of the amount of images that DERETFound used, and about 9% of the amount of images that RETFound-MEH used."

Our expanded experiments further support our original claim that our model is indeed high-performance, and the experiments at a lower resolution show that our approach works well at 224x224, too. For the SSL part especially, we used substantially fewer resources while obtaining a very good end result. While ablating every difference individually (and indeed every combination of differences) would be interesting, it is beyond the scope of our current manuscript. Indeed, we already present a substantial number of results in both the main manuscript and supplementary.

Reviewer #3 (Remarks to the Author):

This paper introduces an efficiency-driven eye foundation vision model. The primary methodology involves a reconstruction pretraining approach rather than a mask prediction pretraining approach. In simple terms, when provided with an image, the model initially employs a frozen image encoder to generate a feature representation. Then, it introduces noise and corrupts patches, using a trainable image encoder to obtain a feature representation (the reconstructed version), which is further compared with the original feature representation. The study claims that this proposed approach demonstrates greater efficiency, requires less training data, and has superior performance compared to two baseline models.

We thank the reviewer taking the time to review our manuscript and for providing their comments, which helped us substantially improve our manuscript. In particular, we have substantially expanded the experimental results in response to the comments by the other reviewers and your own.

In the original manuscript, we compared the three foundation models on their adaptability for downstream classification across two unseen datasets (BRSET and IDRiD) on 29 tasks.

In the revised manuscript, we now compare the three foundation models for:

- Downstream classification across *three* unseen datasets on 43 tasks, adding a Chinese dataset of images of young infants relating to Retinopathy of Prematurity (ROP) and updating our diabetic retinopathy tasks, e.g. considering both the whole of BRSET and BRSET subjects with diabetes only (the latter mirroring a typical diabetic retinopathy screening programme).
- Unsupervised lower-dimensional projections of the feature vectors using both Principal Component Analysis and UMAP for the ROP and IDRiD datasets, qualitatively and quantitatively examining whether the projections (which are obtained entirely without disease labels) separate images by disease status.
- Transportability of feature vectors, using diabetic retinopathy classifiers fit to IDRiD and evaluated on BRSET, and vice versa.

Thus, we not only substantially expanded our original comparisons but added two additional types of comparisons, too. Furthermore, we now also provide results for a RETFound-Green model trained at a lower resolution of 224x224 pixels, denoted as RETFound-Green@224. Finally, we provide additional experiments where relevant in the supplementary.

The new results paint a picture consistent with our original results. RETFound-Green performs the best overall (68 statistically significant wins across 119 comparisons), suggesting that it does not perform systematically worse and possibly generally better than the other two models.

We have made a few minor changes as well. For example, BRSET provides diabetic retinopathy grades using two systems: the Scottish Diabetic Retinopathy Grading Scheme and the International Clinic Diabetic Retinopathy Scale. Both are highly similar, and in the original manuscript we used the Scottish one for no strong reason other than us being based in Scotland. However, IDRiD uses the international scale. Thus, for consistency and to enable the transferability experiments, we now use the international scale throughout. This minorly changed some of the results, but without introducing any meaningful qualitative changes.

Please see our point-by-point replies to your own comments below, and please also consider our responses to the other reviewers where they might be relevant.

Main comments:

The three models (the proposed model and two baselines) are pretrained on entirely different datasets, thus not enabling a direct comparison. At the very least, the study should use the same pretraining dataset and adopt the mask prediction pretraining approach as the baseline to assess performance trade-offs.

We thank the reviewer for this comment. Indeed, if the models were trained on entirely different datasets, this would reduce the directness of the comparison and complicate the interpretation of the results. However, the datasets used for training have substantial overlap which we did not make sufficiently clear in the original manuscript.

In particular, DERETFound used the same three datasets we used (AIROGS, DDR, ODIR) plus an additional dataset (Kaggle EyePACS). For the AIROGS dataset, DERETFound used all available images, whereas we used only 48.8% of the dataset. Overall, our pre-training data is a subset of DERETFound's dataset, with us using one less dataset and slightly less than half of the images the number of images.

For RETFound-MEH, there is no direct overlap with our pre-training data. However, most of the RETFound-MEH pre-training data is from Moorfields Eye Hospital itself and not openly available. Overall, RETFound-MEH used 904,170 images compared to our 75,000 images.

Thus, while we strongly agree with the reviewer in principle that it is important to consider whether differences in pre-training data make the comparison unfair, we do not think that this is a concern in this particular instance. Our pre-training data is a strict subset of DERETFound's, with one less dataset and only half as many images. Thus, DERETFound already represents a baseline of using MAE on an expanded version of our dataset – a more challenging version of what you suggest - and enables such a direct comparison. RETFound-MEH used more than 10 times as many images as we used, which in our opinion makes it very unlikely that our pretraining data gives our model an advantage. On the contrary, we think that having an order of magnitude more data available should generally favour RETFound-MEH.

Based on your feedback, we now make this much clearer in the manuscript. First, we discuss this in the “Datasets for pre-training and downstream evaluations” section:

“We used three publicly available datasets for pre-training: AIROGS [39], DDR [40], and ODIR-2019. We used all images in DDR and ODIR-2019 and then selected a random sample of 53,327 from AIROGS to achieve our desired target amount of exactly 75,000 images. We make the subset of images used available on our GitHub to aid reproducibility. *DERETFound used the same three datasets for pretraining plus an additional dataset and slightly over twice as many images as our model. For a more detailed comparison of the pre-training datasets for the three models, please refer to Supplementary S1.*”

And then clarify this in more detail as follows in the newly added Supplementary section S1:

	RETFound-MEH	DERETFound	RETFound-Green
Datasets	Moorfields Diabetic imAge dataSet (MEH-MIDAS)*, Kaggle EyePACS diabetic retinopathy	AIROGS, DDR, ODIR-2019, Kaggle EyePACS diabetic retinopathy	AIROGS (subsample)*, DDR, ODIR-2019
Notes	* MEH-MIDAS is not openly available.	/	*53,327 randomly selected images, 46.8% of the dataset.
Total images	904,170	150,786	75,000

Supp. Table 1: Overview of the datasets used for pre-training.

“*Supp. Table 1 gives an overview of the datasets used for pre-training for the three models. RETFound-MEH and DERETFound only overlap regarding the Kaggle EyePACS diabetic retinopathy dataset, which can be found at: <https://www.kaggle.com/c/diabetic-retinopathy-detection/data>. The MEH-MIDAS dataset is not openly available.*

DERETFound and RETFound-Green have substantial overlap, with RETFound-Green’s pretraining data being a subset of DERETFound’s. In particular, RETFound-Green used one fewer datasets, and only half of the AIROGS dataset.

Overall, RETFound-Green used slightly less than half of the amount of images that DERETFound used, and about 9% of the amount of images that RETFound-MEH used.”

Furthermore, the study fails to compare with any efficiency-based approaches. The mask prediction pretraining approach, being the default method, does not prioritize efficiency. Numerous efficiency-driven approaches have been proposed, such as pruning-based and distillation-based techniques, which could significantly enhance efficiency compared to the default approach.

We understand this comment to point out that there are methods that could increase the efficiency of existing models, e.g. pruning of model parameters or distilling a larger model into a smaller model. Indeed, such methods could improve the *inference* efficiency of an existing retinal foundation model once trained, i.e. they could make the model slightly more efficient in downstream usage.

However, such methods do not address the initial training costs of the model to be pruned/distilled. Before a model can be pruned or distilled, it needs to be trained in the first place which these types of methods do not address. Indeed, pruning and distillation then require additional computational resources and often lead to reduced performance. While our model being 2.7x faster for inference and 14x smaller in terms of disk space is an additional benefit, the main advantage is that we trained it with 400x less compute (vs RETFound-MEH, 600x less vs DERETFound) and half-the-data (vs DERETFound, 9% vs RETFound-MEH) and achieve high performance.

Furthermore, in our opinion, the implicit assumption in RETFound-MEH and DERETFound is that such resources are required to develop a high-performance retinal foundation model. In a sense, DERETFound does explicitly focus on efficiency – namely data efficiency for pre-training.

At the same time, while we care about efficiency greatly, we would not say that we necessarily “prioritise efficiency” over other factors such as performance. Instead, we deliver high performance and greatly increased training efficiency and increased inference efficiency, all at the same time. This is further supported by our expanded experiments.

We now discuss this as follows in the discussion section:

“RETFound-Green is not only more efficient to train, but also more efficient in downstream usage. In principle existing foundation models, such as RETFound-MEH and DERETFound, could be made more efficient for downstream usage through pruning of less important model parameters or by distilling them into smaller models. However, such approaches would not address the resources required for the initial training. Indeed, they would require additional computational resources and tend to lead to reduced model performance. RETFound-Green is more efficient for downstream usage out of the box while being high-performance, too.”

Another crucial aspect is the limited evaluation of zero/few-shot capabilities. A foundational aspect of a model is its ability to learn a generic representation adaptable to downstream tasks, even for out-of-distribution scenarios. However, the benchmarks predominantly cover diseases included in the pretraining set of the proposed approach. The downstream accountability is not clear.

We thank the reviewer for this very important comment. We completely agree that showing performance across a wider range of tasks, especially diseases that are not present in the pre-training datasets is very important.

Based on your comments, we made three additions to our experiments:

1. We added a dataset relating to Retinopathy of Prematurity (ROP), which we simply call ROP dataset. This dataset contains a disease not present in the pre-training datasets, and is out-of-distribution in another sense by containing fundus images of young

infants, as opposed to the adults in the other datasets. Our model shows very good performance in this dataset.

2. We add unsupervised lower-dimensional projections of the feature vectors, using both PCA and UMAP. These provide a proxy measure of zero/few shot capabilities by highlighting whether and how well the models can identify key axes of variation in the data without labels. We find that our model provides the clearest separation by disease status in an entirely unsupervised way and verify this quantitatively via classifiers fit to the two-dimensional projections and evaluated on held-out test data projected to the same two-dimensional space.
3. We also look at whether classifiers adapted on one downstream dataset transfer to another downstream dataset, using diabetic retinopathy detection and grading in the BRSET and IDRiD datasets as an exemplar. This represents an out-of-distribution scenario for these downstream classifiers fit to the feature vectors of the three foundation models, with the population and imaging devices differing between the two datasets. We find that the classifiers fitted to the original feature vectors transfer well between datasets, with DERETFound and our RETFound-Green getting an equal number of wins. In Supplementary S3, we repeat these experiments for the two-dimensional projections and find that in this case RETFound-Green transfers best.

Thus, these expanded experiments show good performance even in out-of-distribution scenarios, including unseen diseases, and shed light on the performance in unsupervised/few-shot settings.

Additionally, we would like to say that we would like to note that RETFound-MEH was pre-trained on the most data related to diabetic retinopathy – both in relative and absolute terms – so any experiments relating to diabetic retinopathy would (a priori) favour RETFound-MEH over the other two models, including ours. While not exhaustive, in Supplementary S5, we also present a small sensitivity analysis for classification on BRSET, using only 10/20 positive examples per label. As one would expect, performance for all models is worse than when having the whole BRSET training set available. However, the relative performances are consistent with our main findings, with RETFound-Green even obtaining slightly more wins at 10 positive examples per label.

Reviewer #3 (Remarks on code availability):

It only has example codes and does not have pretraining, fine-tuning, and inference codes.

Indeed, the code for pretraining and evaluation is not yet uploaded on Github.

We have attached a zip file with the code for pre-training, image preprocessing, data splits, and evaluation to this response. We kindly ask the reviewers to treat the code confidentially and not use it except for evaluating our work.

We are committed to making all the code available on Github immediately in case our manuscript is accepted for publication.

We would like to note that the example code in the Github is all that is needed for inference of RETFound-Green. The README file explains how to use our model and the model weights are already available in the repository, too, under the release section. So, it is already possible to inference RETFound-Green and to use it for downstream applications.

Reviewer #4 (Remarks to the Author):

Thank you for taking the time to review our work, we really appreciate it. Please see the responses to the comments by the other reviewers, including comments based on your input. These comments helped us to improve our work substantially.

Reviewer #5 (Remarks to the Author):

Thank you for taking the time to review our work, we really appreciate it. Please see the responses to the comments by the other reviewers, including comments based on your input. These comments helped us to improve our work substantially.

REVIEWER COMMENTS

Reviewer #1 (Remarks to the Author):

Thank you to the authors for including a new unseen dataset and additional experiments. However, these additions are not entirely convincing in demonstrating the generalizability of the foundation model without external testing on more unseen datasets. While it is understood that the three unseen datasets were not used for pre-training, external testing remains crucial.

External testing involves fine-tuning the model on one dataset for a specific task and evaluating its performance on the same task using a new dataset from a different cohort, medical center, or ethnic group. Such testing would provide valuable evidence of the model's robustness across diverse data sources and scenarios. Without validation of the foundation model in this setting, it may be difficult to substantiate the claims on the benefits of foundation model.

We thank you for your comment and we are in complete agreement that external testing is critical, especially in healthcare where we need to make sure that models work across different settings, including specific cohort, hospitals, and ethnic groups. Indeed, different cameras used could also affect model performance.

Previously, we had already looked at external testing going from BRSET from Brazil to IDRiD from India and vice versa. We use diabetic retinopathy as it is a very prevalent disease and commonly screened for, with internationally standardized criteria for the disease severity grades. This makes it an excellent exemplar for this external testing. The results we obtained were already promising. We called this “transportability” because we wanted to avoid confusing the reader with “external testing”: All downstream tasks are “external” to the pre-training data, and in these experiments we “transport” an adopted model from one dataset to another.

Based on your feedback that stresses the importance, we decided to expand on these experiments and better signpost what we have done to the reader. Concretely, we add an additional dataset from France called Messidor2. So now the experiments about “transportability” span three different continents, using different cameras and presumably individuals with diverse ethnicity and in differently resourced healthcare settings.

In BRSET, we have patients without diabetes as well, whereas the other two datasets only include diabetics. So as an additional dimension of “distribution shift” between the datasets, we consider either the full BRSET or only patients with diabetes in BRSET. The former represents a more general screening scenario where the patients’ diabetes status is not known, whereas the latter represents a scenario where only patients with known diabetes are invited for screening.

Second, we make this clear to the reader, including by adding the following table to the supplementary S10:

Dataset	BRSET	IDRiD	MESSIDOR2
Country of origin	Brazil	India	France
Continent of origin	South America	Asia	Europe
Cameras used	Nikon NF505 (Nikon, Tokyo, Japan); Canon CR-2 (Canon Inc, Melville, NY, USA)	Kowa VX-10 α (Kowa Company, Ltd., Nagoya, Japan)	Topcon TRC NW6 (Topcon Medical Systems, Oakland, USA)
Diabetes status	Patients with and patients without diabetes	Only patients with diabetes	Only patients with diabetes

Supp?Table 6: Overview of the datasets used for the external transportability experiments.

Third, we renamed “transportability” to “external transportability” to signal that this is external testing, while hopefully avoiding confusion regarding the fact that the other downstream datasets are also external to the pre-training data.

Here are the expanded experimental results:

Figure 5: Performance for diabetic retinopathy related tasks between IDRiD, Messidor2 and BRSET when training the model on one dataset and evaluating on another. Missing bars indicate an AUC < 0.5, i.e. worse than random guessing. For robustness, reported results are the median of 100 bootstrap samples of the test set. The horizontal bars indicate the result of a Wilcoxon signed-rank test between the best and second best methods across the 100 bootstrap samples, with $p < 0.05$ in bold. “***” indicates $p < 0.0001$.

We think that these are very encouraging results that suggest that our model might generalise better to new settings/populations/devices than the two foundation models we compare with. In absolute terms, we also think that these results are encouraging and show that foundation models can indeed generalise during external testing.

For the current work, these results give some support to the generalisability of retinal foundation models and to our claim that RETFound-Green is not systematically inferior to the other two models that were trained with orders of magnitude more compute and more data. Please note that we fully share your view that external testing is crucial and while we think that these experiments are sufficient for the manuscript at hand, we also think that future work

should investigate this in more detail. In fact, we are already starting to work with collaborators across the world to investigate this in detail, including with collaborators in sub-Saharan Africa, which is currently severely underrepresented in our field.

You might also be interested in Supplementary S3 (“External transportability of low-dimensional projections across BRSET and IDRiD”). In the main manuscript, we show that two-dimensional projections of the foundation model feature vectors cluster different diabetic retinopathy stages within a dataset in a fully unsupervised way. In S3, we further show that a 2d projection “fit” to one dataset then also clusters diabetic retinopathy stages in another dataset. In other words, the experiments we added based on your comments show that classifiers using foundation models generalise between very different datasets. S3 further suggests that even the underlying representations in the vector embeddings themselves meaningfully generalise between datasets.

Furthermore, we note that based on the feedback of another reviewer, we have added an additional two datasets in addition to Messidor2 mentioned above. For these datasets, we use the exact same data splits as the original RETFound paper and find results consistent with our previous experiments, further evidencing the strong performance of RETFound-Green. On these three new datasets, we further compare RETFound-MEH after it was fully finetuned (i.e. training the whole model on the downstream task) with RETFound-Green with only linear probing (i.e. fitting a logistic regression to the feature vectors). We find that RETFound-Green with linear probing achieves comparable performance to the fully finetuned RETFound-MEH. This is a very encouraging result as full finetuning requires a lot more resources than simple linear probing.

We thank you again for this comment and we think that the additional experimental results and changes made have substantially improved the manuscript, for which we are most grateful.

Reviewer #2 (Remarks to the Author):

thank you for addressing the issues raised

We again thank you for your feedback which has helped us improve our manuscript substantially.

Reviewer #3 (Remarks to the Author):

Thanks for the substantial efforts on the revision. Many of my comments have been addressed; however, some of my concerns regarding the evaluation remain.

We thank you both for your comments, both the previous and new ones. Just as your previous comments had already helped us improve our manuscript, we now have added important clarifications and additional experiments in response to your points that further strengthened our work.

Briefly, we added three additional datasets – two multi-class ones and an additional diabetic retinopathy one – where we use the same exact splits as in the original RETFound-MEH paper. We further now compare with the fully-finetuned versions of the RETFound-MEH model on this dataset and find that RETFound-Green can achieve comparable performance with only “linear probing”. This is remarkable as full finetuning requires substantially more resources than simple linear probing and yields a new, different model that needs to be stored separately. In the original RETFound-MEH paper, data is split into train, validation and testing sets, with the validation set being used to select the best checkpoint for evaluation. We ignore the validation set for linear probing, meaning that RETFound-Green matches the performance of the finetuned RETFound-MEH checkpoints with not just much less compute, but also with less data.

Finally, we would like to note that we explicitly only claim that based on our results, RETFound-Green is not inferior to the other two models. That is impressive in its own right as our model used orders of magnitude less compute and much less data, while also being more efficient during downstream use. In our opinion, the results we obtain are even suggestive of a slight performance advantage. However, this can differ across tasks and it is unlikely that one model is best in every case (“no free lunch theorem”). We aim to provide comprehensive experiments in this manuscript that support the claims that we make, but future work by ourselves and other researchers should examine performance more closely.

Furthermore, we note that based on the feedback of another reviewer, we have expanded the experiments around “external transportability” where we do linear probing on one downstream dataset and evaluate on another downstream dataset. These results are quite encouraging and suggest that RETFound-Green might have an advantage in terms of generalisability compared to the other two models.

We thank you again for your comments that have allowed us to substantially improve the manuscript. Please find detailed point-by-point replies to your comments below.

The study mentions that the proposed model achieved the best performance in 68 tasks out of 119 comparisons. What exactly is considered a "task"? Are the main classifications in Figure 6 considered one task, with their sub-analyses considered separate tasks?

This indeed was not very clear in the previous version of our manuscript, and we appreciate you pointing that out to us. A comparison refers to each case where we make a statistical comparison between the three models in the main manuscript. Figure 6 (in the old manuscript, in the new manuscript the numbering of figures has changed and this figure has been moved to a supplementary) was not counted in this, because this figure was providing the ablation of RETFound-Green at the same resolution of 224 by 224 pixels as the other two models, rather than RETFound-Green proper that we actually propose.

For example, for diabetic retinopathy grading in a given dataset, we have seven comparisons: one for No/Mild DR vs rest (often referred to as “referrable” vs “non-referrable” DR), one for Any

DR vs no DR, and one comparison for each of the five specific DR grades. The reason why we count all of these as individual comparisons, including the individual DR grades, is that we count the number of statistical comparisons we make. This provides an objective way of tallying up all the results, whereas there are multiple ways that one could aggregate across “sub-“tasks. We use the term “comparison” when counting the number of statistically significant wins and ties. We now clarify this in the relevant methods section:

“We perform a statistical comparison for each predicted label individually and report statistically significant wins and ties at that level, rather than aggregating them across a whole dataset as such aggregation would lose information and would bring additional statistical complexity. The goal of reporting these wins and ties is to give the reader a summary statistic of how the three models compared. Supplementary S9 has tables with detailed breakdowns of the comparisons between the three main models performed in the main manuscript.”

We have added a supplementary S9 with tables that count the comparisons and wins for each model in detail. This makes it clear what exactly is counted and avoids ambiguity.

Figure 2		RETFound-MEH	DERETFound	RETFound-Green
ROP - Retinopathy of Prematurity	Wins	0	0	8
	Ties	0	1	1
BRSET - Diverse tasks	Wins	0	4	5
	Ties	0	0	0
IDRiD - Diabetic Retinopathy	Wins	1	2	4
	Ties	2	2	4
BRSET - Diabetic Retinopathy everyone	Wins	1	0	5
	Ties	0	1	1
BRSET - Diabetic Retinopathy diabetes only	Wins	2	1	3
	Ties	0	1	1
Messidor2 - Diabetic Retinopathy	Wins	1	0	4
	Ties	0	2	2
Retina - Multi-class	Wins	0	1	3
	Ties	0	0	0
JSIEC1000 - Multi-class (non-trivial only)	Wins	2	3	4
	Ties	0	0	0
Total	Wins	7	11	36
	Ties	2	7	9

SuppTable 3: Counts of wins and ties for each subplot of Figure 2.

Figures 3 + 4		RETFound-MEH	DERETFound	RETFound-Green
Retinopathy of Prematurity - PCA	Wins	0	0	11
	Ties	0	1	1
Retinopathy of Prematurity - UMAP	Wins	0	0	12
	Ties	0	0	0
IDRiD - PCA	Wins	0	1	10
	Ties	0	1	1

IDRiD - UMAP	Wins	1	1	9
	Ties	1	1	0
Total	Wins	1	2	42
	Ties	1	3	2

SuppTable 4: Counts of wins and ties for each subplot of Figures 3+4.

Figure 5		RETFound-MEH	DERETFound	RETFound-Green
Messidor2 (France) → IDRiD (India)	Wins	1	0	6
	Ties	0	1	1
BRSET (Brazil) diabetes only → Messidor2 (France)	Wins	0	1	6
	Ties	0	0	0
BRSET (Brazil) everyone → Messidor2 (France)	Wins	0	1	6
	Ties	0	0	0
Messidor2 (France) → BRSET (Brazil) diabetes only	Wins	0	2	4
	Ties	0	1	1
Messidor2 (France) → BRSET (Brazil) everyone	Wins	0	3	4
	Ties	0	0	0
IDRiD (India) → Messidor2 (France)	Wins	2	0	5
	Ties	0	0	0
BRSET (Brazil) diabetes only → IDRiD (India)	Wins	1	1	2
	Ties	2	2	2
BRSET (Brazil) everyone → IDRiD (India)	Wins	1	2	3
	Ties	0	1	1
IDRiD (India) → BRSET (Brazil) diabetes only	Wins	1	3	3
	Ties	0	0	0
IDRiD (India) → BRSET (Brazil) everyone	Wins	0	4	2
	Ties	0	0	0
Total	Wins	6	17	41
	Ties	2	5	5

SuppTable 5: Counts of wins and ties for each subplot of Figure 5.

Ideally, a curious reader would study the results in detail and draw their own conclusions. However, given the numerous experiments and comparisons in our manuscript, we do not expect every reader to examine each experiment in detail. Thus, tallying up the number of wins and ties across all comparisons serves as a coarse but objective way to summarise the results and to support our claim that our model is not systematically inferior to the other two models.

Since the evaluation was conducted on three datasets, it is important to provide details on the exact tasks, including the settings (e.g., train/test or external), and the number of instances.

We fully agree. In response to primarily your own comments as well as the comments by another reviewer, we added another three downstream datasets, which further exacerbates the need for a table summarising the details of each dataset. Thus, we have added the following table to the supplementary S6:

Dataset	Training images	Testing images	Tasks	Data split
ROP	866	233	Retinopathy of prematurity related: ROP grading, classifying ROP vs laser scars vs normal	Random 80-20 split at the patient level
BRSET	13,013	3,253	Retinal abnormalities for three anatomical landmarks (macula, disc, vessels); Binary retinal disease detection (diabetic retinopathy, age-related macular degeneration, macular edema); non-retinal disease tasks (quality scoring, insulin usage, diabetes mellitus); diabetic retinopathy grading and detection	Random 80-20 split at the patient level
IDRiD	413	103	Diabetic retinopathy grading and detection	Official train test split by the dataset creators
Messidor2	972	526	Diabetic retinopathy grading and detection	Same data split as in the original RETFound-MEH paper
Retina	336	181	Multi-class: Normal vs cataract vs glaucoma vs retinal disease	Same data split as in the original RETFound-MEH paper
JSIEC1000	532	318	Multi-class: 39 different classes, we consider 9 classes that are non-trivial as described in the methods section. For more details, please see the original publication by the dataset creators.	Same data split as in the original RETFound-MEH paper

Supp?Table 2: Overview of downstream adaptation datasets.

“The data splits for ROP and BRSET will be released alongside our code upon publication. The data splits from the original RETFound-MEH paper are available here: https://github.com/rmaphoh/RETFound_MAE/blob/main/BENCHMARK.md Note that the data splits from the RETFound-MEH paper include a small validation set (also known as “development” or “tuning” set, distinct from the testing set) that we did not use here. However, the validation set was used by the RETFound-MEH authors to select the best fully finetuned checkpoint, which we compare with in this manuscript. Finally, please also note that due to the relatively large size of BRSET, we additionally provide an ablation in Supplementary S6 of using only small parts of BRSET for adaptation of the foundation models.”

The new datasets are of course also discussed in the relevant methods section where we already discuss the previous datasets.

Additionally, the evaluation is limited compared to the baseline models such as RETFound:

We thank you for your comments in this regard. We have substantially expanded evaluations in response to them. In our opinion, our evaluation is now quite comprehensive and even improves on some limitations the evaluations in the original RETFound paper had. Please find detailed point-by-point replies below.

(1) Only two primary eye disease severity-level classifications were evaluated. Other eye diseases were evaluated as binary classifications (yes/no), which is easier and less clinically useful;

We agree that evaluations should focus on clinically meaningful tasks. In response to this as well as your comment (3) below, we have added two additional datasets that are multi-class with different diseases. We show the results in our reply to that comment below.

Please note that that in the original RETFound paper, for tasks like diabetic retinopathy grading, only a single macro AUC was reported, whereas we report AUCs for each individual grade as well as the two common binary tasks of DR detection (any DR vs no DR) and detection of referable DR (grade 0/1 vs grade 2-4). We include these two additional binary tasks because they reflect common practice in ophthalmology and are thus clinically useful. Thus, our comparison is more fine-grained and less limited compared to the original RETFound paper in this regard.

In addition to diabetic retinopathy, we also consider retinopathy of prematurity and in the case of the IDRiD dataset, we further consider three grades of macular oedema that are provided there. Beyond these three examples, we are primarily limited by the availability of public datasets with more detailed severity grades for other conditions. The reason why we focus on diabetic retinopathy so extensively is in part because it is a very common task in ophthalmology and thus clinically useful, and in part because it is a disease with well-standardised severity grades and publicly available datasets.

Finally, based on your feedback, we now acknowledge this in the manuscript and call for future work to investigate this in more detail.

“While we compare the models across a variety of tasks and find that RETFound-Green generally performs best, it is possible that for other datasets and tasks the relative performance of the three evaluated models could be different. At present, the most important insight is that RETFound-Green does not perform systematically worse while being far more efficient, which is well supported by the data. However, RETFound-Green might represent an improvement in performance and especially the results for the unsupervised low-dimensional projections are very encouraging. Future work should compare the models across more datasets and an even broader selection of tasks. Specifically, this should include classification of disease severity and subtype rather than simple binary tasks as well as progression prediction.”

We completely agree that binarizing more fine-grained labels is almost always unfortunate and we are always disappointed when we come across work that seems to have focused on a binary task to claim higher performance numbers, instead of claiming realistic performance on a clinically meaningful task. We would like to assure you that this is not what we have done here, if we had more fine-grained labels available, we would have gladly used them. However, in our opinion, it is also unlikely that our model performs well and the tasks that we have

considered here, including three instances of severity labels and two multi-class datasets, yet underperforms the other to models in other more fine-grained tasks.

(2) Only classification tasks were evaluated, without including other tasks such as progression prediction;

We agree that this would be very interesting and again we are limited by publicly available data. Based on your feedback, we have now also acknowledged this limitation in the manuscript. Again, we think that given the results we present here, it is unlikely that our model would systematically underperform the other two models on progression prediction.

(See the quoted passage above, “Future work should compare the models across more datasets and an even broader selection of tasks. Specifically, this should include classification of disease severity and subtype rather than simple binary tasks as well as progression prediction.”)

In relation to your comments (1) and (2), we would like to mention that we have used RETFound-Green to win the \$2,000 first prize of the MICCAI2024 Challenge: Structural-Functional Transition in Glaucoma Assessment 2 (STAGE2). This challenge had a complex and clinically meaningful task, namely predicting the results of a visual fields test from imaging. This meant regressing the mean deviation value (task 1), as well as a sensitivity value for each of 52 test points (task 2), as well as a four-way classification of the pattern deviation probability for each of the 52 test points (task 3). The imaging provided was a colour fundus image and a optical coherence tomography volume centred on the macula.

Remarkably, we achieved the overall first place – winning in two of the three tasks and placing third in the last task – by using RETFound-Green vectors of the colour fundus images. 15 teams from good universities participated, and unlike the other teams we did not train a deep learning model on the data and we did not use the optical coherence tomography volumes. In our opinion, this evidences the good performance of RETFound-Green, even for complex tasks and even when competing with excellent data scientists who have a financial incentive to do their best.

We think that the results we present in the manuscript at hand are very strong, but we wanted to mention this to you because it provides strong external validation of the effectiveness of our model. We are unable to include the results of this challenge in the manuscript itself, because the data is not yet available for use outside of the competition and the organisers want to publish a paper on the competition itself, too. Furthermore, our approach for this competition was a bit more complex than the linear probing approach presented here (we considered a global vector plus an average vector for each of four quadrants, then reduced them with PCA and then used a multi-output RandomForest to model co-dependencies) and would require a manuscript of its own to properly explain and ablate.

However, you can verify our claim here:

<https://aistudio.baidu.com/competition/detail/1167/0/leaderboard> The team that is ranked first is from the University of Edinburgh and the team name includes the first name of one of the authors on this manuscript.

(3) Although the study compared its model with RETFound, it did not use any of RETFound's testing sets for direct comparison, nor did it directly cite RETFound's performance.

In our opinion, the comparisons we present are direct and fair. For IDRiD, there is an official train-test-split which we use. For the other datasets, we randomly split the data at the patient-level, use the same splits for all three models, and will make the splits available alongside our code once the manuscript is published.

To address your concerns, we have now added an additional three datasets (Messidor2, "Retina", and JSIEC-1000) to our evaluations where we use the exact same splits as in the RETFound paper. The results on these datasets are consistent with our previous results. In fact, on all three newly added datasets, RETFound-Green achieves the largest number of statistically significant wins.

Note for JSIEC-1000, some of the 39 classes are “trivial” with all models achieving near perfect performance. We focus on a subset of “non-trivial” classes where at least one of the three main models achieves an AUC<0.95, which happen to be 9 of the 39 classes. Full results, including for the trivial classes, are reported in Supplementary S7 for completeness.

Your comment raises another interesting point, namely that we use “linear probing” for our evaluations, i.e. fitting a logistic regression to the feature vectors, whereas the original RETFound paper uses full finetuning. In our opinion, full finetuning has a number of drawbacks

that negate the benefits of a foundation model. First, full finetuning requires a modern GPU workstation whereas linear probing could be done in a few minutes on a laptop. Second, when finetuning the full model, we end up with different weights for the model. So instead of storing and using a single foundation model, we now need to store a separate model for each task. Third, a large benefit of foundation models is the idea that we can have a task-agnostic vector database of image embeddings that we can quickly adapt to new tasks or query with vector-based similarity search. However, when finetuning the whole model, the model would also produce different vector embeddings. Thus, we would need to maintain separate vector embeddings for each task that we finetuned the foundation model for. Fourth, full finetuning has many degrees of freedom in the form of hyperparameters and design choices for the training loop. We would design the training loop differently, but even if then applied our approach to all three models, a reader might wonder if we tuned the training loop to favour our model. Linear probing using scikit-learn default hyperparameters provides a much more objective choice.

To summarize, we think linear probing is most relevant and provides the fairest comparison. In some settings, one might want to do full finetuning, but a good foundation model should provide meaningful vector embeddings out of the box, which is what linear probing tests.

However, on the three datasets where we use the RETFound data splits, we now also compare the fully finetuned RETFound checkpoint as released by the original authors to our model that was only adapted with linear probing. This directly addresses your point of citing RETFound's performance, as the performance reported there is the performance of the fully finetuned model. The results are as follows:

a) Messidor2

b) Retina

c) JSIEC1000 (non-trivial classes)

Figure 6: Comparison of fully finetuned RETFound-MEH using the fine-tuned weights released by the original authors with RETFound-Green with only linear probing. Missing bars indicate an AUC<0.5, i.e. worse than random guessing. For robustness, reported results are the median of 100 bootstrap samples of the test set. The horizontal bars indicate the result of a Wilcoxon signed-rank test between the best and second best methods across the 100 bootstrap samples, with $p<0.05$ in bold. “***” indicates $p<0.0001$.

For Messidor2, RETFound-Green with linear probing has 4 statistically significant wins compared to 3 for the fully finetuned RETFound-MEH. For Retina, both models achieve 2 statistically significant wins each. For the non-trivial classes in JSIEC1000, RETFound-Green with linear probing has 5 statistically significant wins compared to 4 for the fully finetuned RETFound-MEH. We should note that some wins have a large gap between the models, e.g. for optic atrophy the finetuned RETFound-MEH drastically outperforms RETFound-Green with linear probing, while the reverse is true for congenital disc abnormality. Other wins are statistically significant but with a small difference in AUC, e.g. “Any DR” in Messidor2.

On the whole, RETFound-Green with linear probing achieves at least comparable performance to fully finetuned RETFound-MEH. In our opinion, this is an encouraging and impressive result.

Please note that these experiments are intentionally biased against our model. In the RETFound paper, the data was split into training, validation and testing sets. We compare on the same exact testing sets that RETFound used, but only fit our model to the training data. For selecting the best checkpoint, RETFound used the validation set and thus had effectively more data available. We could have added the validation data to the training data or used the

validation data to tune the regularisation strength of the logistic regression, but instead we chose to ignore it entirely.